# Recent Advances in Two-Dimensional MXene for Supercapacitor Applications: Progress, Challenges, and Perspectives

**DOI:** 10.3390/nano13050919

**Published:** 2023-03-01

**Authors:** Zambaga Otgonbayar, Sunhye Yang, Ick-Jun Kim, Won-Chun Oh

**Affiliations:** 1Department of Advanced Materials Science & Engineering, Hanseo University, Seosan-si 356-706, Republic of Korea; zambagaotgonbayar@gmail.com; 2Korea Electrotechnology Reserch Institute, Next Generation Battery Research Center, 12, Jeongiui-gil, Seongsan-gu, Changwon-si 51543, Republic of Korea; shyang@keri.re.kr (S.Y.); ijkim@keri.re.kr (I.-J.K.)

**Keywords:** synthesis method, 2D MXene, electrolyte, MXene-based electrode, supercapacitor

## Abstract

MXene is a type of two-dimensional (2D) transition metal carbide and nitride, and its promising energy storage materials highlight its characteristics of high density, high metal-like conductivity, tunable terminals, and charge storage mechanisms known as pseudo-alternative capacitance. MXenes are a class of 2D materials synthesized by chemical etching of the A element in MAX phases. Since they were first discovered more than 10 years ago, the number of distinct MXenes has grown substantially to include numerous M_n_X_n−1_ (n = 1, 2, 3, 4, or 5), solid solutions (ordered and disordered), and vacancy solids. To date, MXenes used in energy storage system applications have been broadly synthesized, and this paper summarizes the current developments, successes, and challenges of using MXenes in supercapacitors. This paper also reports the synthesis approaches, various compositional issues, material and electrode topology, chemistry, and hybridization of MXene with other active materials. The present study also summarizes MXene’s electrochemical properties, applicability in pliant-structured electrodes, and energy storage capabilities when using aqueous/non-aqueous electrolytes. Finally, we conclude by discussing how to reshape the face of the latest MXene and what to consider when designing the next generation of MXene-based capacitors and supercapacitors.

## 1. Introduction

The types of technology known as “modern electronic devices” have evolved significantly over the past few decades, and they are becoming increasingly important in everyday life [1,2,3,4,5]. The need to create high-quality energy storage accessories has increased dramatically, which can be particularly attributable to the widespread use of home appliances such as wearables and smartphones. Due to their many advantages, such as their fast charging and discharging capabilities and remarkably long life cycles, supercapacitors (SCs)—also known as electrochemical capacitors—are attracting attention for use with various energy storage devices [6,7,8,9,10,11,12]. Since electrode materials are known to be essential for SCs, the development of SCs requires deeper investigation of electrode materials. Electrode materials used for SCs can be divided into electrical double layer capacitors (EDLCs) and pseudocapacitors according to how charges are stored [13,14,15,16]. The most widely used SCs in the market are EDLCs, wherein charges are primarily stored through the revocable electrochemical adsorption/desorption of electrolyte ions at the electrode/electrolyte interface. However, in actual applications, the specific capacitance of EDLCs based on pure carbon materials is limited to 250 F g^−1^ [17]. Meanwhile, pseudocapacitors allow for the storage of charges by a faradaic process involving fast ion intercalation and/or redox reactions near a fast surface/surface. Pseudocapacitive materials often have higher specific capacities than EDLCs because fast surface redox processes can store more charges [18]. The development of SCs has been aided by the use of various pseudocapacitive materials such as electrode materials, including layered double hydroxides (LDHs), transition metal dichalcogenides (TMDs), transition metal oxides (TMOs), and conductive polymers [19]. In addition, porous carbon NCs [20,21] and N-doped porous carbon [22], and the combination of MXene with porous carbon [23], transition-metal compound, and MXene/polymer compounds, have emerged as a promising candidates for SCs [24]. 

These pseudocapacitive materials offer their own advantages, but they suffer from bottlenecks caused by these limitations. By contrast, pseudocapacitive polymers frequently exhibit structural volatility during extraction and interpolation and have short lifecycles when used in SCs. For example, typical pseudocapacitive oxides have low electrical conductivity and require high production costs. Their practicality as SC electrodes is largely limited by this defect [25]. To meet the increasing energy demand of electronic devices and develop new energy storage technologies, it is essential to find new SC-electrode materials that have high strength and stability while also being low-cost. First reported in 2011, MXenes—which consist of atomically thin layers of metal carbides, nitrides, or carbonates—belong to the category of two-dimensional inorganic compounds, and synthesized MXenes consist of various hydrophilic terminations [26]. MXenes are produced from the MAX phase of aluminum, but there have been few reports on the production of MXenes from other A elements [27,28] (e.g., Si and Ga) (Figure 1).

While Ti_3_C_2_T_*x*_ is the most attractive SC electrode material, its performance strongly depends on the demanding properties of MXene, such as surface area, strong hydrophilicity, high electrical and thermal conductivity, and the compatibility of functional groups. SC electrodes based on Ti_3_C_2_T_*x*_ have been shown to have a superior cycle life, superior rate outcome at high scan rates, and relatively high volumetric capacitances. Compared to other forms of MXene, Ti_3_C_2_T_*x*_ may offer a number of benefits when used as a SC electrode. MXene materials used in SCs generally belong mainly to the 2D structure, and from a structural point of view, the horizontal aggregation and restacking of MXene nanosheets due to strong van der Waals interactions between adjacent layers led to the accessibility of electrolyte ions and the use of the entire 2D MXene surface limits these possibilities. One advanced approach to overcome this limitation has been proven to be the design of an open structure of MXene nanosheets, which offers advantages such as increasing the electrochemically accessible surface area of MXenes and improving the rate of ion transport to active redox sites by tailoring the properties or morphology. Various strategies to tune the morphology, such as particle size control, interlayer spacing expansion, three-dimensional (3D) porous structures, and vertical line design, have been proposed in research publications for high-capacity and power-performance MXenes. Designing a 3D/porous electrode structure with a large active surface area accessible to ions and pores connected between ion transport channels can more effectively improve the high-rate capability of MXene. Various methods, such as a hard-templating strategy [29], different types of freeze-drying methods [30,31], chemical-cross linking [32], oxidative etching [33], and self-assembly methods [34], have been used for the preparation of 3D porous MXene materials; most of the 3D porous MXene materials have mesopores/macropores with wide size distribution profiles.

Pliable and wearable electronics are currently undergoing rapid developments in response to the growing demand for wearable technology. SCs are an essential type of energy supply component that ineluctably experience different deformations such as bending, curving, and folding when used in practice. The creation of new SC electrode materials that exhibit exceptional versatility and excellent electrochemical achievement has greatly contributed to the growing energy demands of wearable technology. All things considered, Ti_3_C_2_T_*x*_ has shown remarkable potential as a SC electrode material for real-world energy storage applications. However, Ti_3_C_2_T_*x*_ (MXene) has yet to be extensively examined for use as a SC electrode material, even though there have been many reviews mentioning the benefits of MXenes for electrochemical energy storage. Given the rapid advancements in this area and the advantageous uses of Ti_3_C_2_T_*x*_ (MXene) in attractive pliable/wearable energy storage devices, there is an urgent need for a thorough analysis of SC electrodes based on MXene (Ti_3_C_2_T_*x*_). In order to improve performance and practical applications, it is essential to examine the exact gold properties of MXene materials, including the interaction between MXenes and guest materials and the preservation of physical and mechanical properties of MXenes. Therefore, this paper provides an in-depth discussion of the most recent developments in Ti_3_C_2_T_*x*_-based electrode materials for SCs, and it focuses on the important role played by Ti_3_C_2_T_*x*_ in achieving exceptional electrochemical results, along with related concepts. First, given that the special physicochemical features of Ti_3_C_2_T_*x*_ (MXene) are intimately linked to the production procedures, we will examine the synthesis techniques for Ti_3_C_2_T_*x*_ (MXene) and how they affect the electrochemical properties of the obtained material. The second part will discuss the influence of electrolytes, surface finish groups, and electrode shape on the electrochemical outcome, as well as the suitability of pure Ti_3_C_2_T_*x*_ (MXene) for SC electrode use. Third, since Ti_3_C_2_T_*x*_ composites with materials such as carbon, polymers, and metal compounds have been used to fabricate SC electrodes (Figure 2), the final section discusses the challenges currently facing Ti_3_C_2_T_*x*_-based materials and their potential in SC applications. Altogether, this review paper meticulously discusses the fabrication process, mechanical properties, applications, and related advantages and disadvantages of Ti_3_C_2_T_*x*_-based SC electrodes, as well as the fundamentals of SC design, fabrication, configuration, and application. 

## 2. Synthesis of MXene 2D 

Since the first report of Ti_3_C_2_T_*x*_ MXene in 2011, an ever-increasing number of single and multi-element bidirectional metal carbides, nitrides, and MXenes has been studied. Top-down selective recording is commonly used to generate MXene. This aggregation route has been shown to be scalable without losing or altering properties such as the batch size increases. As shown in Figure 3, many techniques have been used to generate elusive MXenes, including approaches using molten salt [35,36], Lewis acidic molten-salt etching [37], chemical vapor deposition [38,39], electrochemical etching [40], and wet-etching [41,42,43].

For example, the surface of the carbide layers ends with O, OH, and/or F atoms when Ti_3_AlC_2_ is etched in HF solution at environmental temperature, thus resulting in the selective removal of the A(Al) atoms. In addition to HF solvent treatment, recent studies have reported on the use of various reagents and ions, such as DMSO [44], TBAOH [45], and alkali metal ions [46], with sonication treatment methods. Ti_3_C_2_ (MXene) was first reported in 2011, where its synthesis was described by immersing Ti_3_AlC_2_ (MAX) powder in a 50% concentrated HF solution (in Figure 4a). The final MXene exhibits an accordion-like layered structure with multiple functional groups. Since Mxene was first reported, the forms of MXene have come to include multifarious Ti_2_CT_*x*_, Ta_4_C_3_, TiNbC, (V0.5,Cr0.5)_3_C_2_, and Ti_3_CN using different etching times and different HF concentrated solutions [47]. The below equations express the overall chemical reactions of the etching process:M_n+1_AlX_n_ + 3HF = AlF_3_ + 3/2H_2_ + M_n+1_X_n_(1)
M_n+1_X_n_ + 2H_2_O = M_n+1_X_n_(OH)_2_ + H_2_(2)
M_n+1_X_n_ + 2HF = M_n+1_X_n_F_2_ + H_2_(3)

Moreover, single- or few-layer Ti_3_C_2_T_*x*_ can be obtained by mixing with DMSO molecules after preparing Ti_3_C_2_T_*x*_ using an HF solution, then using ultrasound to increase the interlayer spacing or loosen the interlayer interactions [42]. An exfoliated Ti_3_C_2_T_*x*_ prepared by HF etching of Ti_3_AlC_2_ as a SC electrode material was first reported in 2013 (Figure 4b) [48]. Multilayer Ti_3_C_2_T_*x*_ was also decomposed using DMSO to form fewer layers. Subsequently, MXene particles were fabricated as filter-free paper electrodes [48]. However, DMSO has been shown to be an effective cross-linking agent for delayering Ti_3_C_2_T_*x*_ [25] and (Mo_2/3_Ti_1/3_)_3_C_2_T_*x*_ [49]. As has been shown for V_2_CT_*x*_ and Ti_3_CNT_*x*_, overlapping samples can also be prepared using other organic molecules such as TBAOH [46]. In addition to HF strong etching agent treatment, the non-direct use of HF etching treatment on MAX was reported in 2014 by Gogotsi and Barsoum [46]. Along with the LiF/HCl mixture solvent, fluoride salts/HCl mixture solvents such as NaF, KF, NH_4_F, and FeF_3_/HCl as well as fluorine-containing ammonium salts have been used for the preparation of Ti_3_C_2_T*_x_*, Ti_2_CT*_x_*, and V_2_CT*_x_*. An advantage of this method is that the etching exfoliation process proceeds in one step, and the entire synthesis path reduces the high usage of the HF agent. The interlayer distance of MXene is also increased due to the intercalation of Li^+^ and NH_4_^+^ ions. Therefore, the products only require manual or ultrasonic stirring to be delaminated directly into monolayer flakes. However, because monolayer MXene is easily oxidized in both air and aqueous solutions, methods to prevent the oxidation and storage of MXene at low temperatures, such as the use of additives—which do not interfere with the use of MXene—have been reviewed. All of these methods have excessively used F-based etching agents using HF or F-containing etching agents, which leads to considerable safety and environmental problems. In attempts to resolve these major problems, research on the use of fluorine-free etching solutions for the development of MXene is gradually becoming increasingly popular. In 2018, an F-free hydrothermal alkali etching method was used for the synthesis of MXenes [50]. Therefore, MXene formation from MAX (Ti_3_SiC_2_) with non-Al was reported using a hybrid HF/H_2_O_2_ etchant solution [51]. Since then, different synthesis studies have been reported in 2019, such as adopting MXene metal oxides, MXene polymers, and MXene–carbon NCs, representing the composite MXene as a state-of-the-art hybrid material for versatile applications [52,53,54]. In 2016, Urbankowski et al. proposed a short period and effective etching method by using a molten fluorine-containing salt mixture of LiF, NaF, and KF as an etchant to obtain the first two-dimensional transition metal nitride MXene [36]. As shown in Figure 4d, the use of Lewis acid molten salt in the preparation of MXene has been reported with various MAX and A elements, such as Zn (Ti_3_ZnC_2_), Al (Ti_2_AlC), Si (Ti_3_SiC_2_), and Ga (Ti_2_GaC). This etching technique offers remarkable opportunities to tune the surface chemistry and assets of MXenes, along with the potential to generate new MXenes, including those that cannot be prepared using the etching process. In addition to fluorine-containing salt melts, Lewis acid melt salts have proven to be effective tools for etching the MAX phase. The “A” atom in the MAX precursor, and the cation in the Lewis acid, are removed by an oxidation-reduction reaction. In 2020, Talapin et al. proposed a general strategy and synthesized a variety of MXenes with single surface terminations (O, S, Se, Cl, Br, NH, and Te), as well as bare MXene without surface termination, and found that some of the Nb_2_C series MXenes, including Nb_2_CCl MXene, have superconductivity under low-temperature conditions [55].

The abovementioned etching methods were based on removing the “A” layer from MAX and forming MXene with and without surface terminations, and the lateral dimension of the chemically derived MXene was between several hundred nanometers to ∼10 μm. Three different syntheses—chemical vapor deposition (CVD), lithiation expansion−micro explosion mechanism, and in situ electrochemical synthesis methods—are listed in the bottom-up approach used for MXene synthesis. In 2015, Xu et al. proposed a CVD method for synthesizing ultrathin and large-scale MXenes (Mo_2_C) [39]. Two years later, another research group reported a CVD method for MXene, which had a lateral dimension at the centimeter level [56]. In 2020, Buke et al. studied the influence of impurities on Mo_2_C crystal formation [57]. For the lithiation expansion–micro explosion mechanism, Sun et al. developed a new method for preparing single-layer or few-layer MXenes from Ti_3_AlC_2_ MAX. In addition to that, the Ti_3_SiC_2_ MAX was used as a precursor material in this method (in 2019) [58]. Recently, in 2020, Zhi et al. reported an integrated process that combines the in situ etching of MAX and ion storage of MXene using LiTFSI and Zn(OTF)_2_-mixed ionic electrolyte as the etchant solution [59]. 

## 3. Key Properties of 2D MXene for Supercapacitor

The 2D MXene family members have outstanding properties that can be tuned by controlling the stoichiometric ratios of the M and X elements. Subsequently, the unique and tunable properties of MXene—including good mechanical properties, dispersibility in water, hydrophilic nature, metallic-like conductivity, and electrochemical active surface—make them potential candidate materials for the design of high-power and high-energy density capacitors. The unique properties of MXene are sequentially and concisely summarized in the following subsections. 

### 3.1. Excellent Mechanical Flexibility of MXene

Two-dimensional-structured MXene exhibits excellent mechanical properties, which make the MXene highly suitable for use in pliant supercapacitor devices. The mechanical properties of the material includes its strength, elasticity, tractability, and machinability, and among these, the flexibility properties of the material allow it to form in any shape. Recent years have brought significantly increased demand for compact and holdable “micro-electronic” system devices, such as pliable SCs and micro-SCs, and the malleable and 2D-lamellar structure of MXenes make them promising candidates for such devices. Two-dimensional MXene nanosheets can be easily assembled as film electrodes [60,61,62], and most of the research to date has focused on further improving the structural properties of MXene (as shown in Figure 5). The current research results show that progress is being made in these directions.

### 3.2. Hydrophilic and Dispersibility of Mxene

The hydrophilic surface of MXene has a strong affinity for water, and the spreading of water over such surfaces is preferred. However, the ability of electrode materials to disperse in a variety of solvents is crucial for creating electrode inks that are subsequently used to create electrodes or composite electrodes. In the preparation of MXene, the choice of synthesis methods offers the control of the surface terminus (-OH, -F, -Cl, -O); thus, the –O and –OH groups are important in conferring the hydrophilicity of MXene in a stable manner in aqueous solution (in Figure 6). Moreover, the electrochemical outcome of the MXene-based electrodes is primarily driven by the surface terminus groups of MXene [63].

### 3.3. Conductivity of MXene

Supercapacitors typically have a high energy density and fast discharge rate; therefore, they require electrodes with high conductivity, similar to Me, and the advantages of MXene in facilitating the achievement of the above characteristics include the selection of M and X and the adjustment of their surface termination groups to improve the conductivity of MXene, which can be turned into metal-like, semi, or full insulation [64,65]. Therefore, the large lateral widths and low defect concentrations in MXenes are often associated with higher conductivities [66,67]. As shown in Figure 7, fiber-molded Ti_3_C_2_T_*x*_-MXene film etched against cold-pressed Ti_3_C_2_T_*x*_ (MXene) film (1000 S cm^−1^) has severe defects. By contrast, spin-cast Ti_3_C_2_T_*x*_ (MXene) film can obtain an extremely high conductivity (6500 S cm^−1^) after being etched with a LiF/HCl solution [68,69]. Obtaining large power densities requires the very high conductivity of MXenes, but it can also reduce the need for conductive materials to make electrodes or even current collectors [47], thus boosting the device’s overall energy density.

### 3.4. Charge Storage Mechanism

The pseudocapacitive ion-interstitial charge storage mechanism of MXenes contributes to their high capacity and long life. Longevity and capacity are key components of a SC. The 2D layered structure of MXene allows for the rapid intercalation of ions (e.g., H+, Li+, or Na+), thus enabling both rapid pseudocapacitive charge storage as well as efficient operation in acidic aqueous electrolytes and (as has recently been demonstrated) in electrolyte-free media. As a result, MXenes have high capacitance (300–500 F g^−1^) or capacitance more or less comparable to that of EDLC carbon materials in aqueous electrolytes; further, over thousands of cyclic operations, MXenes outperform pseudocapacitive materials with little or no capacitance loss. In addition, the capacitance of MXene-based electrodes in SCs can be enhanced by controlling the composition of MXene-containing NCs. For example, the Ni-MOF/Ti_3_C_2_T_*x*_ nanocomposite has a high specific capacitance of 497.6 F g^−1^ at 0.5 A g^−1^ in KOH/PVA electrolytes [70], Ti_3_C_2_T_*x*_-AuNPs film has capacitance of 696.67 F g^−1^ at 5 mVs^−1^ in 3 M H_2_SO_4_ electrolytes [71], and MnO_2_@V_2_C-MXene has a capacitance approximately 551.8 F g^−1^, and a retentivity of about 96.5% after 5000 cycles [72]. These results reveal an efficient approach to fabricating high-performance metal oxide-based symmetric capacitors can be proposed, and a simple and easy way to improve the design of MXene-based electrodes is presented.

### 3.5. High Density and Gravimetric Capacitance of MXene

High-volume MXenes produce high-density, high-gravity MXenes. In practical applications, the increase in device power and energy density is highly dependent upon the volume capacitance of the material [45,67]; available materials include vacuum-filtered Ti_3_C_2_T_*x*_ (MXene) membrane, which can reach a volume capacitance of 900 F cm^−3^, while MXene hydrogel membrane can attain a volume capacitance of 1500 F cm^−3^ [73]. Figure 8 shows all the data collected. These results are superior to those that have been obtained for other supercapacitor electrode materials, such as the best-activated graphene electrode (200–350 F cm^−3^) or porous AC (60–200 F cm^−3^) [74,75], while achieving the same outcomes.

## 4. Designing 2D-MXene Electrode

A battery and an electrochemical capacitor are two examples of energy storage technologies that have long been used to bind ions together into layered materials, and few core materials contain ions that are relatively larger than lithium. As has already been mentioned, MXenes exhibit quick ion intercalation behavior, thus leading to a wide active surface that is accessible for electrochemistry and suitable for use as a pseudocapacitive mechanism in acidic aqueous and non-aqueous electrolytes. The first report on MXene in a condenser electrode was published in 2013 [47]. This study demonstrated that cations of various saline solutions were spontaneously intercalated between the MXene layers. Moreover, the MXenes mix a hydrophilic—mainly hydroxyl-terminated—surface with 2D conductive carbide layers. Thus, it is true that the electrochemical interaction of many cations such as Na^+^, K^+^, NH_4_^+^, Mg^2+^, and Al^3+^ can provide a capacitance of more than 300 farads per cubic centimeter (which is relatively higher than that of porous carbon). Therefore, in this study, not only were MXene (Ti_3_C_2_T_*x*_) flakes studied, but Ti_3_C_2_T_*x*_ binder-free paper was also prepared and used as a capacitor electrode. According to their study, a variety of cations of different charges and sizes from aqueous solutions can be successfully intercalated on exfoliated Ti_3_C_2_T_*x*_-multilayers and on MXene paper consisting of several Ti_3_C_2_T_*x*_ layers. Both the pH and cation makeup affect this phenomenon. A thorough examination of the Ti_3_C_2_T_*x*_’s electrochemical characteristics in these aqueous electrolytes revealed its significant intercalation capacitances. In summary, the first report of MXenes in capacitors with mono- and multivalent ions provides a basis for exploring the 2D superfamily for their suitability in electrochemical energy storage applications. The study presented above used the HF solvent as an etching agent for Ti_3_C_2_T_*x*_ synthesis, while another research study used the mixture of LiF/HCl as an etching agent. According to their study, the LiF/HCl-etched Ti_3_C_2_ (MXene) clay exhibited outstanding 245 F g^−1^ gravimetric capacitance with an infinitesimal capacitance loss after 10,000 cycles at 10A g^−1^. Moreover, while utilizing hydrogel Ti_3_C_2_T_*x*_ (MXene) electrodes in 3M H_2_SO_4_ electrolyte, the use of glassy carbon current collectors led to the achievement of capacitance as high as 380 F g^−1^ or 1500 F cm^−3^ [76]. F-containing agents other than the LiF/HCl and HF etching agents were used; however, the energy storage of the MXene-based electrode did not outperform that of the above-mentioned results.

Figure 9 shows the different patterning methods that are used for MXene-based SCs. Generally, the fabrication methods include two classes: (a) direct patterning of MXene on the current collector/substrate by laser scribing or reactive ion-etching; (b) transferring the MXene ink into different patterns by following printing methods. In addition to these methods, various types of methods have been used for the preparation of MXene-based SCs, such as spray coating with laser cutting, vacuum filtration with laser cutting, direct writing, and freeze-drying with laser cutting methods.

## 5. Charge Storage and Its Influence on the Surface Group of MXene

The energy storage process of MXene involves three main processes: electron transfer to the surface, ion propagation from the electrolyte to the inter-layer, and electron collection in the collector. Therefore, the conductivity of MXene-based electrodes should be as high as possible to store energy at high rates. The TMC core formed by the M and C layers interspersed in one layer of M_3_C_2_ MXene is responsible for the conduction of electrons. The better electrical conductivity of Ti_3_C_2_T_*x*_ than that of Ti_2_CT_*x*_, as has been reported experimentally, may be explained by the lack of a carbide-core in an M_2_C MXene. The second metal replaces the outer layer of the first metal in ordered double TM-MXenes, thus forming the structures M1_2_M_2_C_2_ and M1_2_M2_2_C_3_, where M_1_ and M_2_ represent two separate initial transition metals, which may be Mo, Cr, Ta, Nb, V, or Ti (as shown in Figure 10) [77]. 

Ti_3_C_2_T_*x*_, made by extracting additional aluminum from the Ti_3_AlC_2_ (MAX) phase, has the highest electrical conductivity of any known MXene—20,000 S cm^−1^. As the outermost layer becomes Mo, the metal-like characteristics of MXene change from metal to semiconductor, EDLC can be developed on a surface of MXene, and pseudocapacitive ions can be intercalated through the surface oxidation-reduction reaction. As illustrated in Figure 11a, the interacting ions interact with MXene surface groups to partially transfer their electrons to a material, therefore causing a surface oxidation reaction on MXene. Thus, the charge storage mode of MXene electrodes is the pseudocapacity mode, and directly dependent upon this quality, MXene can maintain and transmit substantially more capacity than EDCL capacitors. To evaluate the effect of surface modification on the electrochemical properties, Ti_3_C_2_T_*x*_ powder mixed with MXene powder 80 wt%, 15 wt% carbon black, and 5 wt% PTFE binder was processed into individual electrode films and tested in 3M H_2_SO_4_ aqueous electrolyte. As shown in Figure 11b, a nitrogen surface functionalization has enabled access to the full potential of MXene in a highly reversible, high-rate-performing method. It would be interesting to extend these measurements to higher voltage windows, as even higher voltages cannot drive more ions to electro-adsorb in the current mode; that is, we expect the curve to bend down at higher voltages. 

Due to the surface oxidation-reduction charging mode, the surface chemical of MXene affects the capacity of charge storage MXene-based electrodes. A functional group is added to the surface of MXene after etching the MAX-A layer in a fascia solution. The treatment of different etching agents has been shown to generate various electrochemical characteristics, and the Ti_3_C_2_X (X = Cl, Br.) has been shown to manifest different improved capacity. Moreover, surface defects and voids of MXene act as active sites for ion adsorption and other interfacial reactions, which can affect rate-speed capabilities, potential windows, and the capacitance of materials. In many cases, Ti-deficient defects are clustered in the same sublayer (Figure 11c–e), and the concentration of surface defects can be adjusted by changing the amount of etching agent used in the etching process. Figure 11c shows the typical defect configurations in a single-layer Ti_3_C_2_T_*x*_ flake: Ti vacancy (VTi) and Ti vacancy clusters, with clusters ranging from 2–17 VTi. The defect concentration was tuned using an HF etchant with different concentrations, which influenced the Ti vacancies on the surface functional groups, thereby adjusting the electronic conductivity. The ability to tune the defect concentration is critical to control the functional properties of the MXene phases. The microscopic crystal structures were further characterized by high-resolution TEM (HRTEM) and the images are shown in Figure 11d. 

## 6. MXenes as Electrode Materials for Supercapacitors

The synthesis method of MXene shows various surface termini and topographies with different energy storage properties, and there have been multiple studies examining surface modification, stoichiometric ratio, and electrode composition control. The following subsections summarize this research.

### 6.1. Control of Size of MXene Flake

The morphology and structure of MXenes are strongly influenced by the corrosivity of the synthesis conditions. For example, prolonging the etching time produces a more open structure than the HF etching of Ti_3_AlC_2_, which produces an accordion-like Ti_3_C_2_T_*x*_ (MXene). Ti_3_C_2_ with a corrosion time of 216 h has a deformed octahedral structure that loses more Ti and C contact on the surface than Ti_3_C_2_ with a short corrosion time of 24 h, thus giving it significantly better capacitive achievement [81,82], as shown in Figure 12. However, despite the higher ionic conductivity and easier access to ion diffusion pathways, the smaller MXene fragments have higher electrical conductivity due to the higher resistance to intersurface contact. Kayali et al. [83] examined how the side diameters of MXenes affect their electrochemical properties. In addition, Mustafa et al. [71] demonstrated that rational design electrodes are fabricated by mixing MXene with an aqueous solution of chloroauric acid (HAuCl_4_). As shown in Figure 12g,h, the etching time had an influential effect on electrochemical properties of MXene. The symmetric supercapacitors made of MXene and AuNPs have shown exceptional specific capacitance of 696.67 F g^−1^ at 5 mV s^−1^ in 3 M H_2_SO_4_ electrolyte, and they can sustain 90% of their original capacitance for 5000 cycles (in Figure 13). The highest energy and power density of this device, which operates within a 1.2 V potential window, are 138.4 Wh kg^−1^ and 2076 W kg^−1^, respectively. From topography analysis, it is clearly shown that Au NPS are homogeneously located on a MXene nanosheet; therefore, the Au NPs served as the best intercalators in the hybrid structure [84]. The thickness of the MXene/AuNPs composite film had stacked layer with a 12 μm thickness.

### 6.2. Category of Composition

Table 1 summarizes most of the previous studies examining Ti_3_C_2_T_*x*_ and other MXenes for energy storage applications or supercapacitor electrodes. Ti_2_CT_*x*_ MXenes [85,86] deliver outstanding outcomes. Compared to carbides, nitride-based MXene exhibits higher electrical conductivity [87]. Accordion-structured, -O/-OH-terminated Ti_2_NTx nanolayers are obtained by synthesizing the Ti_2_AlN (MAX) phase by oxygen-assisted molten salt followed by HCl treatment. 

Nanolayered Ti_2_NT_*x*_ MXene with -O/-OH surface terminations exhibited a capacitance of 200 F g^−1^ at a scan rate of 2 mV s^−1^ in 1 M MgSO_4_ electrolyte, and this MXene material was prepared by HCl treatment on Ti_2_AlN-MAX phase. In addition to traditional Ti-based MXene materials, members of the MXene family are expanded indefinitely by using M-centered metal alloys to include C- and N-bonded transition metal species, such as Ta- [93] and Mo- [42] based MXenes, which are valued as reliable electrodes for supercapacitor applications. Conductivity, stability, and electrochemical outcomes are all significantly affected by the intrinsic properties of the M and X elements, and the surface finishes significantly affect these qualities as well. On the other hand, since their properties can be modified by options of different M and X components and surface terminals can be managed, there is tremendous potential to develop new MXene with favorable capacity properties. Therefore, considering their potential utility in supercapacitors, newer MXenes with higher electrochemical efficiency are needed. An appropriate synthesis approach, including etching and separation processes, should be optimized for each MAX precursor because it has a unique atomic bond. This is necessary to achieve the desired structure and achievement. There is a need for a large amount of research to investigate new MXene and preparation techniques, which represents an important and difficult task.

### 6.3. Heteroatoms Doping and the Control of Surface-Terminus Group

The surface properties and electron acceptability of the material can be changed by heteroatom doping. N-atom has been concluded to be an active heteroatom that affects the electrochemistry of MXenes, and many studies have examined these relationships. The reaction mechanisms can be summarized as follows: by replacing the C-atoms in the C-layer with N-atoms, it has been demonstrated that the distance between layers increases, effectively improving the electrical conductivity [94]. The most commonly used N-doping methods can be classified into heat treatment in ammonia [92], doping in liquid intermediates, water heat treatment [93,95], solvent heat treatment [96], and liquid peeling [97]. For example, as illustrated in Figure 14, a Ti_3_C_2_ film with a flexible N-doping layer was prepared by static adsorption and in situ solvent heat treatment using an alcohol solution saturated with urea as a source of N-atoms [96]. In addition, the intercalation of N in MXene increased the distanced of layers, therefore it offers the opportunity to tune the electrochemical properties. 

Moreover, Ti_2_CT_*x*_-MXene underwent simultaneous liquid-gas phase delamination and chemical doping with nitrogen [97]. Under inert environs, the NH_2_CN utilized as a N-source and intercalant was bonded to the Ti_2_CT_*x*_ nanosheets, and during heat treatment, a condensation reaction resulted in polymeric carbon nitride (p-C_3_N_4_). Solid-state doping was also shown to be possible. N and S co-doped Ti_3_C_2_T_*x*_-MXene with a 75 F g^−1^ at 2 mV s^−1^ in Li_2_SO_4_ electrolyte were produced by pyrolysis of thiourea [98,99]. Nitrogen has been the primary focus of the majority of experiments investigating MXene doping. The potential to alter MXenes has also been demonstrated by using other types of doping atoms, such as metals and heteroatoms. The calculation results show that B-doping improves the stretchability of MXenes, thus making them ideal for use in pliable-electronics [100]. The interlayer spacing, conductivity, and capacitance of MXenes can also be increased by adding metal atoms such as V [101] and Nb [102] doping. As discussed above, chemical modification can be used to fine-tune the properties of MXenes and improve their outcome rate. As vacancies can lead to flaws in the MXene matrix and subsequently affect the structure, surface properties, and activity of MXenes, it may also be a useful strategy for modifying their properties. Novel types of terminus and doped atoms have been added to MXenes as a result of the growing depth of research examining chemical modification techniques [103,104,105,106,107].

### 6.4. Fabricating Vertical Alignments

The RGO/Ti_3_C_2_T_*x*_ MXene-modified fabrics were fabricated by a facile dip-coating and spray-coating method and the schematic illustration of the procedure is shown in Figure 15. The electrochemical performance of RGO and MXene-modified fabrics were characterized via CV tests in two-electrode configuration by using the 1 M H_2_SO_4_/PVA gel electrolyte. Figure 15a shows the CV curves of all-solid-state MCF and RMC supercapacitors at a scan rate of 10 mV s^−1^. The MCF-based supercapacitor shows the smallest enclosed CV curve (almost a line), indicating its inferior charge storage capability. Meanwhile, the charge storage capability was significantly increased after the incorporation of RGO, as the RMC-1 shows an enlarged CV area. In addition, the incorporation of RGO significantly enhanced the C_A_ to 62.9 mF cm^−2^ for RMC-1, and the C_A_ was further increased to 92.5 mF cm^−2^, 115.0 mF cm^−2^, and 258.0 mF cm^−2^ for RMC-2, RMC-3, and RMC-4, respectively. The increased C_A_ was ascribed to the synergistic effects between the increased electrical conductivity and dominated pseudocapacitance of MXene. The topography images of the RGO/Ti_3_C_2_T_*x*_ MXene were analyzed by SEM and the obtained images are shown in Figure 15g,h. As shown in the SEM image, the RGO and MXene sheets are compactly filled in the fiber gaps and fiber surfaces, and graphene sheets can be obviously observed in the gaps between the weft yarns and warp yarns.

Efficient longitudinal ion transfer can be achieved by fabricating MXene membrane electrodes from “T-shaped” fragments [108]. The Ti_3_C_2_Tx film electrode in the H_2_SO_4_ electrolyte showed a capacity ranging from 2 A g^−1^ to 361 F g^−1^, and showed a capacity increase of 76% when the current density was increased to 20 A g^−1^. Table 2 summarizes the design and results of electrodes that can meet the capacity and current efficiency requirements of a supercapacitor. As illustrated in Table 2, the theoretical limit of MXene capacity can be approached by designing a hydrogel framework, while high-speed results can be obtained at a scan rate of up to 10 V s^−1^ using the porous MXene structure, and high capacity power with a film thickness of up to 200 μm can be obtained using vertically oriented MXene particles. This can confirm the output result. These designs and their combinations have provided new and exciting possibilities for MXene in supercapacitor applications.

**Figure 15 nanomaterials-13-00919-f015:**
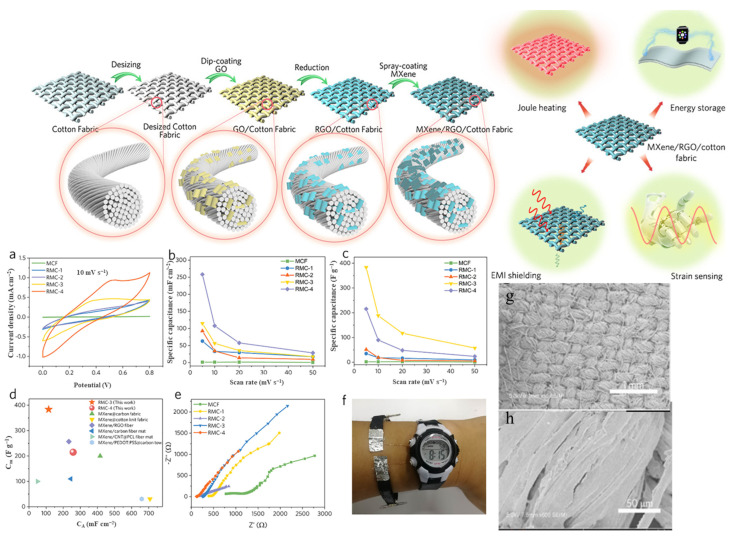
Schematic illustration of multifunctional RGO/Ti_3_C_2_T_*x*_ MXene fabrics, and electrochemical performance of the MCF and RMC fabrics. (**a**) CV curves of the MCF and RMC at 10 mV s^−1^. Areal specific capacitance (**b**) and gravimetric specific capacitance (**c**) of MCF and RMC. (**d**) Areal and gravimetric specific capacitance of RMC compared with the reported MXene-modified fibers, yarns, and fabrics. (**e**) Nyquist plots of MCF and RMC. (**f**) Four RMC-4 supercapacitors connected in series as a wristband to power a watch. (**g**,**h**) SEM images of fabrics [109].

**Table 2 nanomaterials-13-00919-t002:** Capacitive outcome of MXene electrodes with optimized structural design.

Electrode	Preparation	Electrolyte	Capacitance	Cyclability	Ref.
Controlling flake size					
Ti_3_C_2_T_*x*_ film	Mixing large and small flakes	3 M H_2_SO_4_	435~86 F g^−1^	10,000 cycles	[69]
Adding spacer between MXene interlayer					
75 μm Ti_3_C_2_T*_x_*pillared by hydrazine	Suspending in hydrazine	1 M H_2_SO_4_	250~210 F g^−1^	no decay (10,000 cycles)	[110]
Ti_3_C_2_T*_x_*/graphene 3% film	Mixing and filtration	3 M H_2_SO_4_	438~302 F g^−1^	no decay	[111]
Sandwiched Ti_3_C_2_T*_x_*/CNT 5% film	Alternative filtration	1 M MgSO_4_	390~280 F cm^−3^	no decay	[112]
V_2_CT*_x_*/alkali metal cations film	Cation-driven assembly	3 M H_2_SO_4_	1315~>300 F cm^−3^	106 cycles	[113]
T_i3_C_2_T*_x_*ionogel film	Immersing into EMITFSI	EMITFSI	70~52.5 F g^−1^	1000 cycles	[114]
Ti_3_C_2_/CNTs film	Electrophoretic deposition	6 M KOH	134~55 F g^−1^	no decay	[115]
carbon-intercalated Ti_3_C_2_T*_x_*	In situ carbonization	1 M H_2_SO_4_	364.3~193.3 F g^−1^	10,000 cycles	[116]
Ti_3_C_2_T*_x_*@rGO film	Plasma exfoliation	PVA/H_2_SO_4_	54~35 mF cm^−2^	1000 cycles	[117]
Designing 3D/porous structure					
13 μm Ti_3_C_2_T*_x_* film	1~2 μm PMMA template	3 M H_2_SO_4_	310~100 F g^−1^	-	[59]
MXene/CNTs film	Ice template	3 M H_2_SO_4_	375~92 F g^−1^	10,000 cycles	[118]
Ti_3_C_2_ aerogel	assembly and freeze-drying	1 M KOH	87.1~66.7 F g^−1^	10,000 cycles	[119]
Ti_3_C_2_T*_x_*hydrogel	assembly and freeze-drying	H_2_SO_4_	370~165 F g^−1^	10,000 cycles	[54]
Ti_3_C_2_T*_x_*/carbon cloth	Freeze-drying with KOH treatment	1 M H_2_SO_4_	312~200 mF cm^−2^	8000 cycles	[120]
Ti_3_C_2_T*_x_*/Ni foam	Electrophoretic deposition	1 M KOH	140~110 F g^−1^	10,000 cycles	[121]
Compact and nanoporous Ti_3_C_2_T*_x_* film	Freeze-drying and mechanically pressing	3 M H_2_SO_4_	932~462 F cm^−3^	5000 cycles	[122]
3D porous Ti3C_2_T*_x_* film	Reduced-repulsion freeze casting assembly	3 M H_2_SO_4_	358.8~207.9 F g^−1^	10,000 cycles	[123]
Fabricating vertical alignments					
Anti-T Ti_3_C_2_T*_x_* film	Filtrating through an entwined metal mesh	1 M H_2_SO_4_	361~275 F g^−1^	10,000 cycles	[106]
200 μm Ti_3_C_2_T*_x_* film	Mechanical shearing of a discotic lamellar liquid-crystal MXene	3 M H_2_SO_4_	>200 F g^−1^	20,000 cycles	[105]

### 6.5. 3D Microporous Sphere/Tube Ti_3_C_2_T_*x*_

Due to strong van der Waals interactions, both V- and Ti-based MXenes tend to re-stack, resulting in decreased surface area and potassium ion kinetics, and many methods have been developed to fabricate 3D architectures using PICs in attempts to solve this problem. To generate 3D microporous spheres or tubes made of Ti_3_C_2_T_*x*_, Fang et al. devised a facile spray-lyophilization process [124]. Figure 16a,b shows a 3D Ti_3_C_2_ structure consisting of several smooth-surfaced spheres and some tubes. The development of 3D morphology successfully solved the stacking problem of MXene nanosheets and enabled the possibility of fast ion transport (Figure 16c). The constructed PICs offered significant energy density and power density (98.4 Wh kg^−1^ and 7015.7 W kg^−1^). According to Figure 16d,e, Zhao et al. 3D K-pre-intercalated Ti_3_C_2_T_*x*_ (MXene) (K-Ti_3_C_2_T_*x*_) was synthesized by electrostatic flocculation, freeze-drying, and KOH treatment, and the MXene interlayer gap escalated to 1.32 nm after K-intercalation, which was advantageous for the K^+^ kinetics. The proposed PICs exhibited an energy density of 163 Wh kg^−1^ and a high power density of 8.7 kW kg^−1^ with the combination of an AC cathode with a 3D K-Ti_3_C_2_T_*x*_ anode. In addition to the above study, the Zhang et al. research group fabricated a 3D foam-like 3D FMS by electrostatically neutralizing Ti_3_C_2_T_*x*_ with positively charged melamine followed by calcination [115], where the K conductivity was accelerated due to a surface area of 89.5 m^2^ g^−1^. Figure 16f shows the schematic illustration of the synthesis procedure of the 3D-FMS nanocomposites. 

### 6.6. Design of 3D Porous MXene Electrode

By assembling 2D MXene wafers into a 3D electrode structure, the achievement of MXene materials for energy storage applications can be improved. The design of the MXene 3D foam was completed using a variety of technologies. Therefore, it is advantageous to reduce the size of the fragment and increase the interlayer distance so that the transport and accessibility of ions to the active site can be improved to increase the efficiency of MXene. However, by forming a 3D/porous electrode structure with large active surfaces that are accessible to ions as well as interconnected holes for ion-transport channels, the high throughput potential of MXene can be increased further. The achievement of MXene is expected to increase as appropriate porous structures can provide paths for electrolyte wetting and ion transport, thus reducing electrical resistance and shortening diffusion distance. Porous MXene electrodes have been generated by a variety of methods, such as chemical etching [33], template method [29], hydrazine-induced foaming [128], and electrical deposition [129]. Recently, small ice particles have been used as a self-sacrificing template for forming a flexible MXene/CNT film having large pores and nanopores. The controlled porous structure allows for capacities up to 375 and 251 F g^−1^ at 5 and 1000 mV s^−1^, respectively. Foaming, corrugation, and carbon assembly have also been suggested to be effective ways to create porous MXene frameworks [130,131,132,133].

It is also possible to combine 3D MXene aerogels with graphene [115]. As shown in Figure 17, both GO/MXene nanosheets are forced to gradually arrange along the ice crystal boundary during the lyophilization of the mixed GO and MXene solution, and they are eventually crosslinked through interactions to create a 3D lattice porous structure.

## 7. MXene Composite Material for Capacitor Electrode

MXenes have been considered to be potential building blocks for composites for use in energy storage applications due to their distinctive 2D wafer structure and superior electrical conductivity. MXenes have been combined with multiple active ingredients, including metal oxides and conductive polymers, to produce a synergistic effect. However, these elements increase interlayer spacing and reduce MXene rearrangements, which is expected to improve ion accessibility and accelerate ion transfer. Therefore, MXene-based composites often exhibit enhanced electrochemical capabilities, thus indicating the potential utility of MXene in supercapacitor technology. The studied MXene-composite materials and their electrochemical properties in the capacitor field are reviewed in the following subsections.

### 7.1. MXene/Conducting Polymers

Organic polymers, also known as conductive polymers, or more accurately, intrinsically conducting polymers (ICPs), are those that conduct electricity. The substances in question may be semiconductors or have metallic conductivity. Charge transfer complexes were the first highly conductive organic compounds, because they were the first conducting polymers, such as polyaniline, which was first identified in the middle of the 19th century. Superconductivity was first demonstrated in 1980; however, researchers first showed that salts of tetrathiafulvalene exhibit conductivity that is virtually metallic in the early 1970s. Conducting polymers have significant benefits for wearable supercapacitors compared to other pseudocapacitive materials due to their inherent flexibility and conductivity. The composites of MXene and conducting polymers can also achieve high capacitance in acidic solutions. Multiple conducting polymers, including PANI [135], PEDOT [136,137], PPy [138], PDA [139], and PFDs [140], have been combined with MXenes to produce hybrids with excellent electrochemical characteristics.

A conducting polymer that is commonly combined with MXene is the highly conductive PANI [135,141,142], and CP@rGO electrodes with Ti_3_C_2_T*_x_* film electrodes have been formed by polymerization as shown in Figure 18a [135]. Figure 18b shows the strategy to expand the voltage window of the CP-containing supercapacitors by pairing them with 2D Ti_3_C_2_T*_X_* sheets in 3 M H_2_SO_4_. The topography image of the MXene electrode films confirmed that MXene film is composed of well-aligned stacked sheets, as shown in Figure 18c. Figure 18d–i shows the CV curves (5 mV s^−1^) of the individual electrodes (Ti_3_C_2_T*_x_*, and CP@rGO) in a three-electrode setup in which both electrodes showed redox activity at entirely different potentials, demonstrating the possibility to manufacture all-pseudocapacitive asymmetric devices in aqueous electrolyte. The formation of Ti_3_C_2_T*_x_*//PEDOT@rGO showed it had stable pseudocapacitive character even at a scan rate of 100 mV s^−1^, confirming facile proton transport within hybrid devices.

### 7.2. MXene/Metal and MXene/Metal Oxides

Pseudocapacitive materials for supercapacitors include a variety of metal-based compounds. These materials often display below-average electronic conductivity despite having high theoretical potential. Due to the metal-like conductivity of MXene, composites consisting of MXene and metal have been investigated as a potential solution to this problem. For MO, MnO_2_, MoO_3_, RuO_2_, SnO_2_, TiO_2_, NiO, WO_3_, and ZnO are often used in combination with MXene to obtain excellent capacitance [143,144,145,146]. MXene/MO composites can be generated by in situ processing or field assembly. A simple technique to induce meaningful interactions between components for in situ synthesis is to chemically grow metal oxides directly on liquid phase MXene nanosheets [147]. Figure 19a shows the synthesis procedure of CoS_2_ nanoparticles’ growth on a MXene surface and a simple one-step solvent thermal method has been used for the synthesis.

From SEM images, it can be observed from the figure that the composite has two kinds of morphologies: one is spherical structure and the other is lamellar structure. A synergistic effect can be generated between them, promoting its electrochemical performance. Figure 19b–g represents the electrochemical tests of the MXene/CoS_2_ (CCH) composite in 2 M KOH aqueous electrolyte. As show in Figure 19b–g, the MXene/CoS_2_ (CCH)/rGO ASCs can reach a maximum operating voltage window of 1.6 V. In addition, the specific capacitance retention of the device is about 98% after 5000 cycles of charging and discharging, meaning it has excellent long-term cycling performance. It is worth noting that the capacitance drops slowly in the initial 1000 cycles and increases to 98% in the range of 1000–5000 cycles. The possible explanation is the activation of electrode materials and the remarkable enhancement of wettability between active materials and electrolyte ions.

TiO_2_/Ti_3_C_2_ composites can be produced in situ by adding TiO_2_ precursors or by directly oxidizing Ti_3_C, because TiO_2_ is a metal oxide that can be made by converting Ti_3_C_2_ MXenes. Using tetrabutyl titanate as a precursor, TiO_2_ NPs were incorporated into Ti_3_C_2_ nanolayers, where in situ hydrolysis resulted in a high specific surface area, large interlayer distances of MXene, and the transport path of MXene open ions [146]. Then, for the synthesis of Ti_3_C_2_/TiO_2_ nanocomposites, a less difficult one-step oxidation process at room temperature was proposed (Figure 20).

Under various hydrothermal conditions, Ti_3_C_2_ MXene has been coupled with monoclinic WO_3_ nanorods and hexagonal WO_3_ NPs using HCl and HNO_3_, respectively [149]. The capacitance of the hexagonal WO_3_/Ti_3_C_2_ composites (566 F g^−1^) was nearly twice that of pure hexagonal WO_3_ in 0.5 M H_2_SO_4_ electrolyte within a −0.5 to 0 V vs. Ag/AgCl potential range (Figure 21). A microwave process has also been used to combine metal oxides and MXenes [150]. MXenes in composites in basic electrolytes primarily facilitate rapid electron transfer and sporadic flexibility with minimal capacitance contribution. MXenes also serve as active materials in acidic electrolytes because they can offer high pseudocapacitance based on redox reactions starting from H^+^ intercalation.

## 8. Capacitive Mechanism of MXene in Electrolytes

The development of new or improved methods for MXene-based electrodes for energy storage applications and their achievement of high capacity are based on the basic understanding of the energy storage mechanism of MXenes. Moreover, the electrochemical signature of MXene in acidic electrolyte under a high scan rate environment shows broad redox peaks on top of a rectangular CV shape, which is indicative of redox pseudocapacitance. Intercalation pseudocapacitance is a storage mechanism characterized by the location of H^+^ insertion reaction sites between layers of MXene in an acidic electrolyte without phase transition [151]. This property contrasts with the sluggish ion intercalation that is observed in battery-type layered electrode materials such as graphite, and it also has a long cycle life (more than 10,000 cycles). This section reviews and discusses recent developments in the field of research examining the capacitive properties and charge storage processes of MXenes in various electrolytes. For capacitors, various types of electrolytes have been used, including acidic, aqueous (basic and neutral), and non-aqueous electrolytes. Because acidic electrolytes use a different charge storage mechanism to basic and neutral electrolytes, MXene electrodes have the maximum capacitance in H_2_SO_4_ electrolytes in acidic electrolytes, along with excellent rate production. Figure 22a [152] compares the CV profiles of Ti_3_C_2_T_*x*_ film electrodes in acidic and neutral electrolytes. The existence of pseudocapacitive behavior can be seen in the H_2_SO_4_ electrolyte by looking at the large redox bumps at −0.4 to −0.3 V in the CV curves. In contrast to its neutral equivalents ((NH_4_)_2_SO_4_ and MgSO_4_), the H_2_SO_4_ electrolyte produces a substantially larger capacitance [152]. The results of in situ X-ray absorption spectrometry (XAS) measurements reveal the change in the oxidation state of Ti during the electrochemical discharge activity of Ti_3_C_2_T_*x*_ in H_2_SO_4_, thus confirming the pseudocapacitive behavior based on redox reactions [106]. Figure 22b uses in situ electrochemical Raman spectroscopy to represent the discharge process of the Ti_3_C_2_T_*x*_ film electrode in the H_2_SO_4_. According to [152], a reversible voltage-subordinate change is observed after discharge, which indicates an electrochemical reaction:(M=O*x*) + 1/2*x*H+ + 1/2*x*e- → M-O(1/2)*x* (OH)(1/2)*x*(4)

In contrast to basic and neutral electrolytes, acidic electrolytes exhibit pseudocapacitive behavior with redox reactions, thus leading to higher capacitance and faster kinetics as a result of rapid proton transport. For example, they can continue to deliver 500 F cm^−3^ at a potential scan rate of 1000 mV s^−1^ [70]. There is much evidence showing that acidic electrolytes and electrode materials that exhibit rapid faradic reactions, such as metal oxides, are compatible [14]. In acidic electrolytes, the higher capacitance of MXene (more than 1500 F cm^−3^) can be fully utilized. However, despite the fact that MXenes may function with high capacitance and rate in acidic aqueous electrolytes based on the intercalation pseudocapacitive process, the actual utility of such devices is constrained by their limited operating voltage range (approximately 1 V).

To attain high capacity, water molecules must be present between the MXene layers. They support ion intercalation and adsorption [153,154,155] as well as electrical double layer (EDL) formation in the MXene interstitial space. As shown in Figure 23, hydrated cations, such as Li^+^, are formed by water molecules around the cation intercalated in MXene layers without dehydration. The hydration shell isolates the atomic orbitals of the cation and the MXene, prevents orbital hybridization, and separates the positive and negative charges. Due to this potential difference in the spacing between layers, an internal EDCL is generated. Thus, despite the intercalation of hydrated cations in the interstitial space, MXenes exhibit most of the properties typical of an EDLC with a near-constant capacity in the potential window of the aqueous electrolyte.

Solid-state electrolyte (SSE)-based SCs have recently attracted considerable interest due to the rapidly increasing power demands of wearable electronics, portable electronics, printable electronics, microelectronics, and highly flexible electronic devices. Electrolytes are essential components of supercapacitors and play an influential role in transferring and balancing the charge between the two electrodes. It is true that the interaction between electrolyte and electrode in electrochemical processes broadly affects the state of the electrode–electrolyte interface and the internal structure of the active material, and it is noteworthy for the advanced application of flexible supercapacitors. Designing MXene materials and suitable solid electrolyte/MXene interfaces is a highly valued approach. SCs composed of aqueous electrolyte ionic liquids are available to operate at high voltage and possess high conductivity and capacitance, but have a leakage problem. The use of SSEs may help to avoid the leakage problem [156]. SSEs are composed of inorganic-solid and organic-polymer electrolytes. The inorganic-solid electrolytes include a single crystal, amorphous compound and are polycrystalline, whereas organic-polymer solid electrolytes are composed of an organic polymer matrix and a salt. Most SSEs for SCs have been based on polymer, including solid polymers, gel polymers (quasi-solid state), and polyelectrolytes. Among these three SSEs, gel-polymer electrolytes have been extensively used in SCs due to their high ionic conductivity derived from the liquid phase. For the preparation of gel-polymer electrolytes, various types of polymer used include PVA, PMMA, PVDF-HFP, etc., and water, some organic solvents (DMF, EC, PC) are used as a plasticizer [157,158,159,160]. Figure 24 is a graphical representation of a hydrogel polymer composed of a polymer host (PEO, PVA or PEG) and an aqueous electrolyte (H_2_SO_4_, KOH. etc.,) or a conducing salt dissolved in a solvent [161].

The comparison study results of the pure MXene SCs, MXene composite SCs, and other SCs are summarized in Table 3. As shown in Table 3, MXene-based SCs have long-term reusability properties with high gravimetric capacitance, which further strongly confirms the theoretical reason that MXene has emerged as a candidate for SCs.

## 9. Conclusions

It is of utmost importance to review the exact gold properties of MXenes materials in terms of improving performance and practical applications. In addition, it is challenging to fabricate finely structured films reliably and inexpensively, and further investigation of the communication and interactions between MXenes and guest materials is considered as a further area of review. In addition to the material composition, the design of new advanced MXene-based SCs by improving the electrochemical properties while maintaining mechanical properties such as flexibility, strength, and hardness of the material is an influential area of future research.

The key properties of MXene, such as its 2D slab structure, metal-like conductivity, density, variable surface endpoints, and pseudo-interactive capacitance, make it a good candidate. In this paper, we describe the latest advances in MXene-based electrodes for energy storage applications and review recent studies on the synthesis technology, structural design, chemical changes, electrochemical characterization, and basic knowledge of charge storage mechanisms in various electrolytes. Methods used in the construction and electrochemical capabilities of MXene-based hybrid supercapacitors and other devices are also discussed. Since their discovery in 2011, MXenes have been prepared using three main synthetic methods: HF etching with subsequent cation intercalation and delamination, etching in moderately F-containing solutions (such as LiF/HCl or NH4HF2), and etching in F-free electrolytes (such as Lewis acidic molten salts). F-free preparation is considered a desirable approach in terms of electrochemical performance, safety, and environmental friendliness. Ti_3_C_2_T_*x*_ is the subject of most research reported in the literature, with less focus on alternative compositions. Energy storage is one of the key applications that can leverage the diversity of MXene materials that can be generated and their ability to modify surface chemistry. A fundamental research plan will need to be established to develop our basic knowledge of the MXene/electrolyte interface and to develop new methods to control surface chemistry. Even if the 2D lamellar structure of MXene nanosheets improves the ion accessibility and the resulting electrochemical reaction rate, it is important to prevent agglomeration and restacking. To address this problem, various structural and electrode architecture approaches have been developed, such as generating vertical alignment and controlling the flake size and interlayer spacing. In particular, in terms of speed capability, the display of MXene is greatly enhanced by rational shape and structural design. One of the most important techniques for improving the electrochemical properties of MXenes is surface chemical modification. MXene can perform chemical transformation through doping and surface adjustment. When a material is manufactured using a fluorinated corrosive electrolyte, an additional manufacturing method without fluorine as well as a post-treatment process to remove the F-terminal can be developed, thus disrupting payload transport and reducing the number of active sites. Thus, the selection of an electrolyte is also a major influencing factor on the electrochemical results of MXene electrodes, as the improvement of the electrochemical result of MXene depends on the MXene/electrolyte interface and the mastery of the charge storage mechanism in non-aqueous electrolytes. Furthermore, reinforced materials can be incorporated into MXene matrices without affecting the electrochemical results of flexible and printable MXene-based supercapacitors to improve their mechanical properties. The infinite possibility of generating new MXenes and tuning their functions suggests that advances in MXenes for SC are still in their early stages despite numerous improvements and achievements. Understanding capacitive energy storage mechanisms, managing surface chemistry, and creating designs that support high-performance supercapacitors are the next steps.

Due to their unique physicochemical properties and typical structural features, MXenes can offer many new potential applications. To date, various approaches to overcome the problems associated with MXene continue to be reported in the field of energy storage, advancing the design process for improving the MXene layer and enabling further potential applications related to electromagnetic shielding, sensors, and wearable SCs. Continued rapid development of fundamental concepts and technical developments related to MXenes can be expected to open the door to more exciting discoveries.

## Figures and Tables

**Figure 1 nanomaterials-13-00919-f001:**
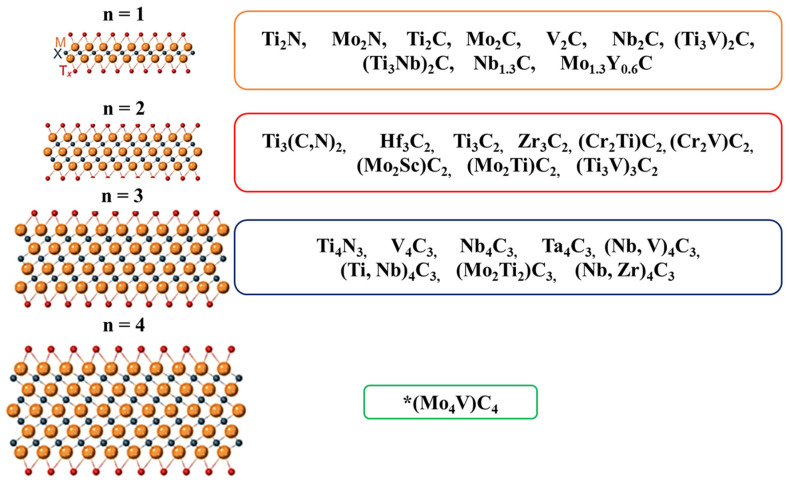
The list of studied MAX and MXene nanocomposites.

**Figure 2 nanomaterials-13-00919-f002:**
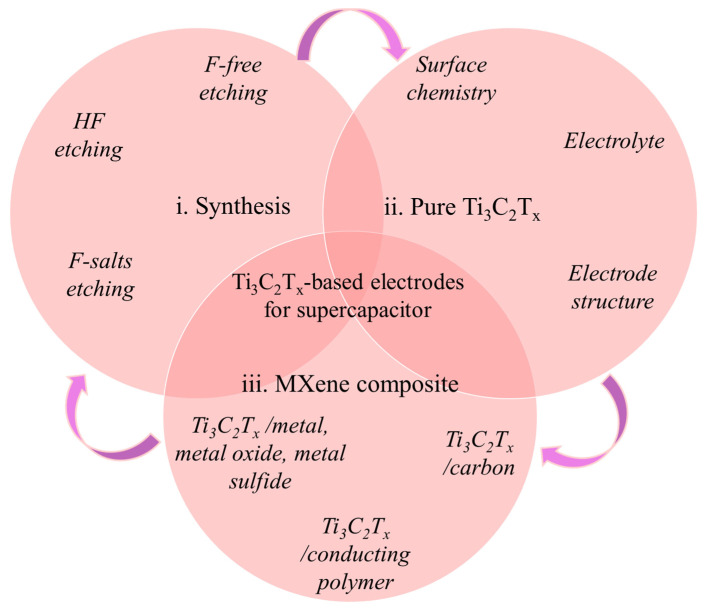
Overview of typical synthesis, variables affecting electrochemical achievement, and Ti_3_C_2_T_*x*_-based composite electrodes for SCs.

**Figure 3 nanomaterials-13-00919-f003:**
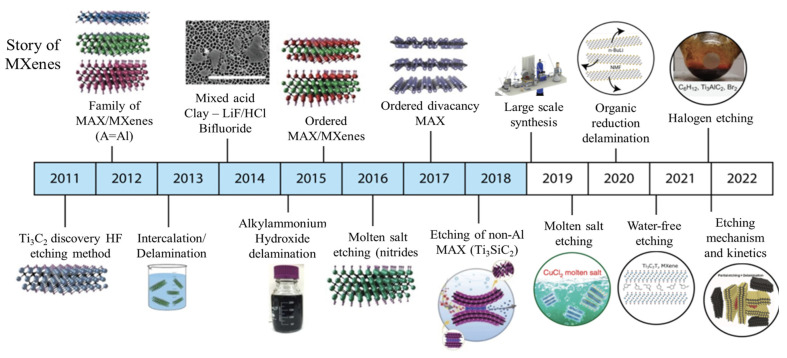
Synthesis methods of MXene from 2011–2022 [35,36,37,38,39,40,41,42,43].

**Figure 4 nanomaterials-13-00919-f004:**
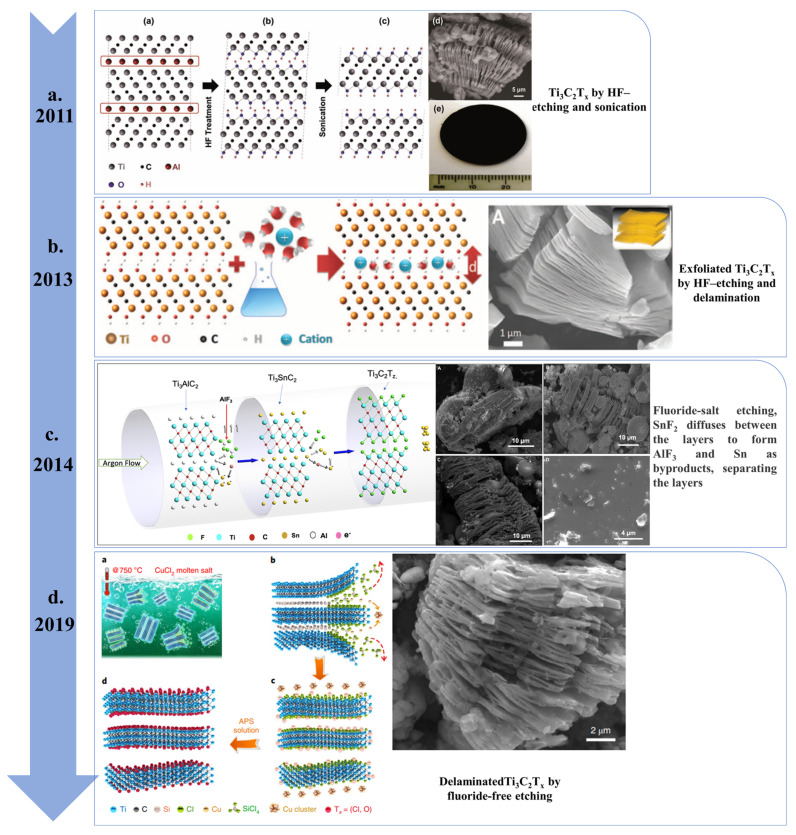
(**a**) Ti_3_C_2_T_*x*_-MXene [46], (**b**) MXenes-based electrode for SCs [47], (**c**) Ti_3_C_2_T_*x*_-MXene clay [45], and (**d**) Ti_3_C_2_T_*x*_-MXene by molten-salt [37].

**Figure 5 nanomaterials-13-00919-f005:**
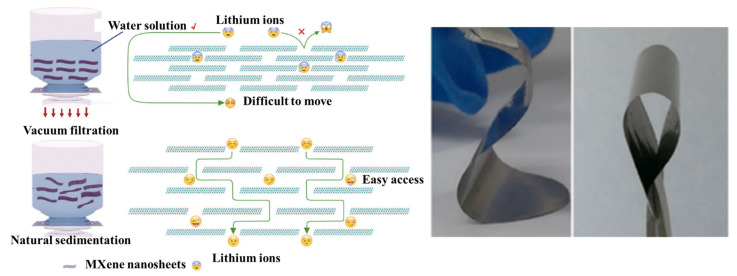
Comparison of enhanced ion availability for digital photography of naturally deposited MXene films [60].

**Figure 6 nanomaterials-13-00919-f006:**
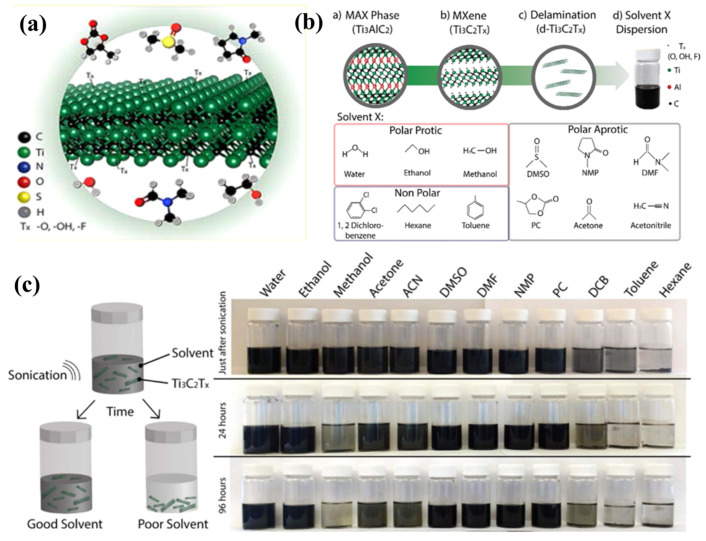
(**a**) An exemplification of MXene with (**b**) description of the synthesis and dispersion of Ti_3_C_2_T_*x*_ in organic solvent, (**c**) and dispersibility of Ti_3_C_2_T_*x*_ in 12 different solvents [63].

**Figure 7 nanomaterials-13-00919-f007:**
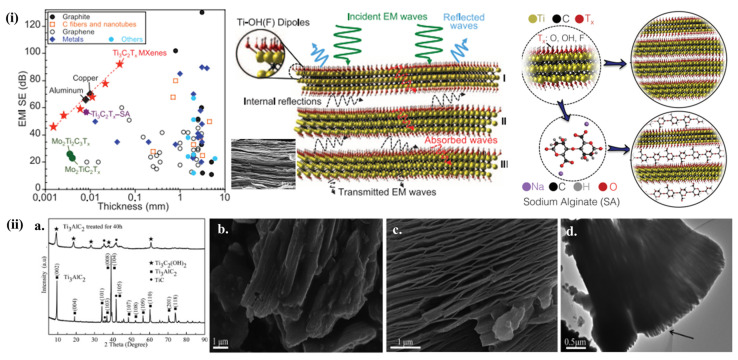
(**i**) An exemplification of TC and TC-SA composite with EMI SE, (**ii**) (**a**) XRD, and (**b**–**d**) SEM/TEM images before and after HF treatment on MAX [68].

**Figure 8 nanomaterials-13-00919-f008:**
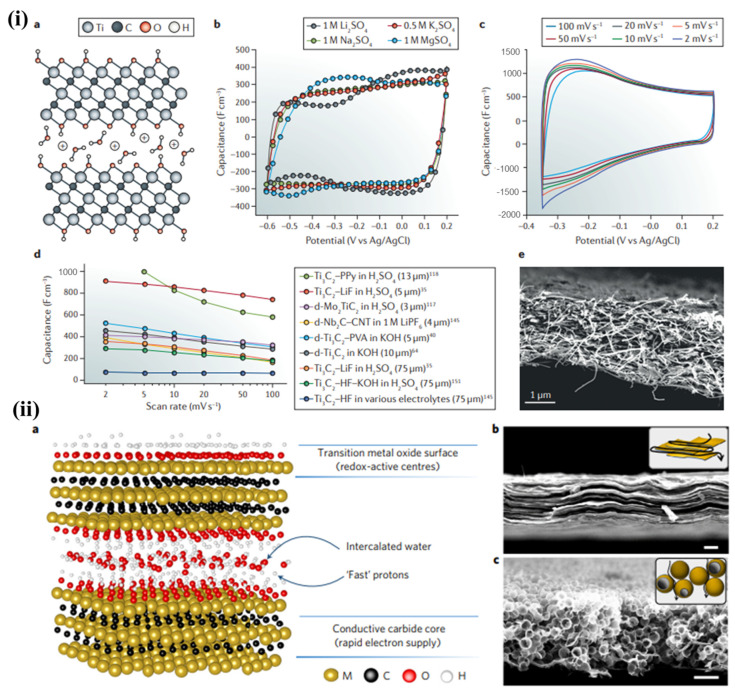
(**i**) An exemplification of MXenes. (**b**,**c**) Cyclic voltammograms of Ti_3_C_2_T_*x*_ paper electrode. (**d**) Comparison study of MXene electrodes. (**e**) Cross-sectional image [67]. (**ii**) An exemplification and SEM images of Ti_3_C_2_T_*x*_ MXene [74].

**Figure 9 nanomaterials-13-00919-f009:**
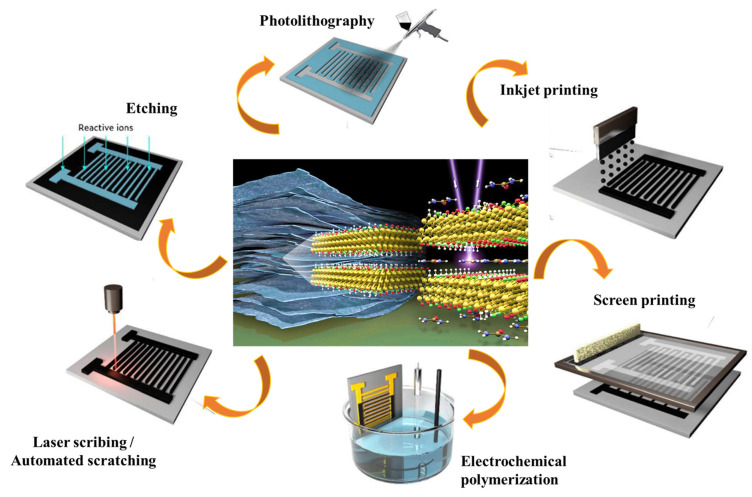
The fabrication methods for interdigital MXene SCs.

**Figure 10 nanomaterials-13-00919-f010:**
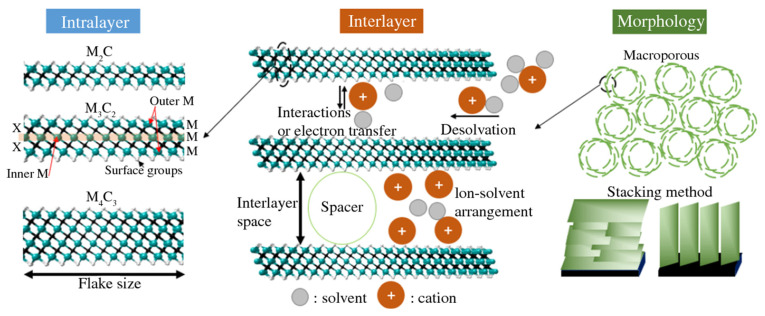
Rate determination of 2D materials at three levels: intra and inter-layer, and electrode morphology [77].

**Figure 11 nanomaterials-13-00919-f011:**
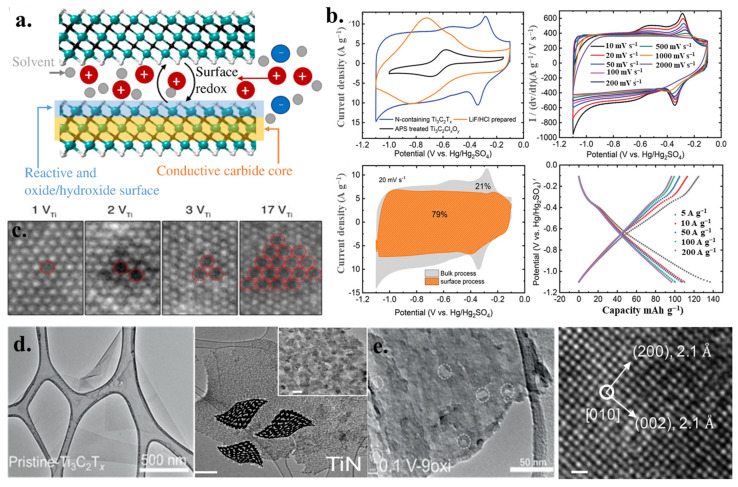
(**a**) An exemplification of MXene. (**b**) Charge storage rate [78]. (**c**) Dark-filed TEM [79]. (**d**,**e**) TEM image TM-nitride nanocrystals at 700 °C [80,81].

**Figure 12 nanomaterials-13-00919-f012:**
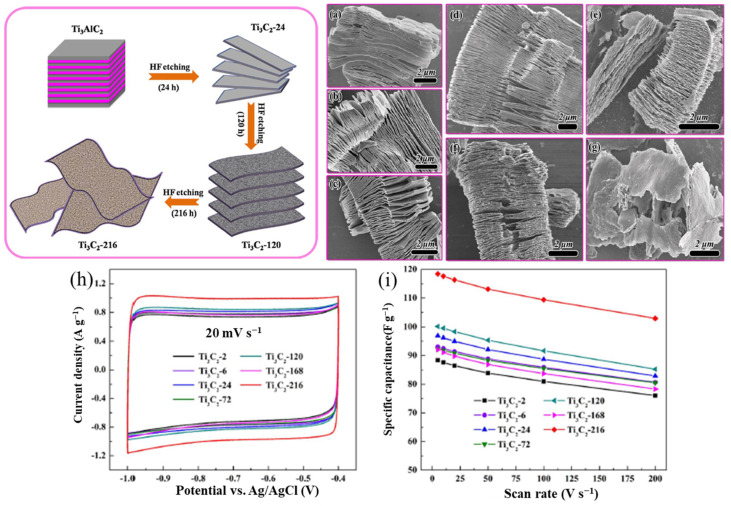
Schematic illustration of the typical structures of Ti_3_C_2_ with the evolution of the etching time: SEM images of the typical as-fabricated MXene with the CV curves and specific capacitance vs. scan rate for the electrodes with different etching times [81,82].

**Figure 13 nanomaterials-13-00919-f013:**
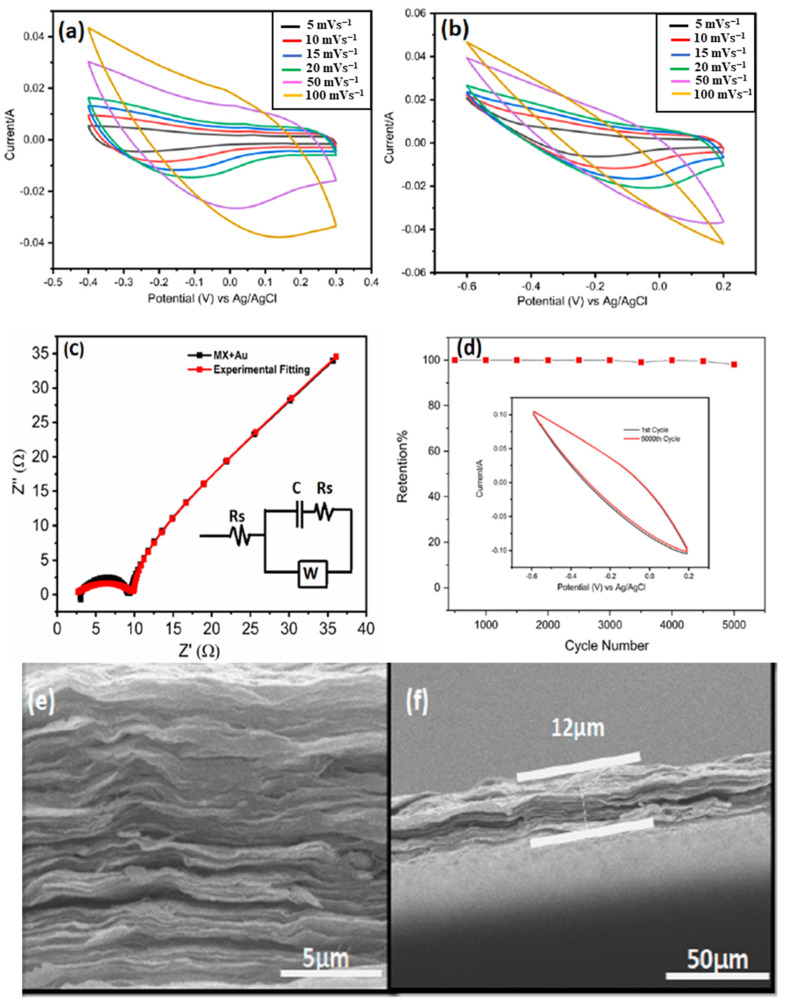
(**a**) CV curves of MXene at different scan rates in 3-electrode setup; (**b**) CV curves of MX/AuNPs in 1 M H_2_SO_4_ at different scan rates in 3-electrode setup; (**c**) Nyquist plots of MX/AuNPs with inset showing the equivalent circuit model for the Nyquist plots; (**d**) MX/AuNPs electrode showing excellent cyclic stability with 98% capacitance retention at 100 mV s^−1^ 1M H_2_SO_4_ after 5000 cycles; (**e**) cross-sectional image of the MX/AuNPs composite film demonstrating stacked layered structure; (**f**) cross-sectional image showing the thickness of the composite film [84].

**Figure 14 nanomaterials-13-00919-f014:**
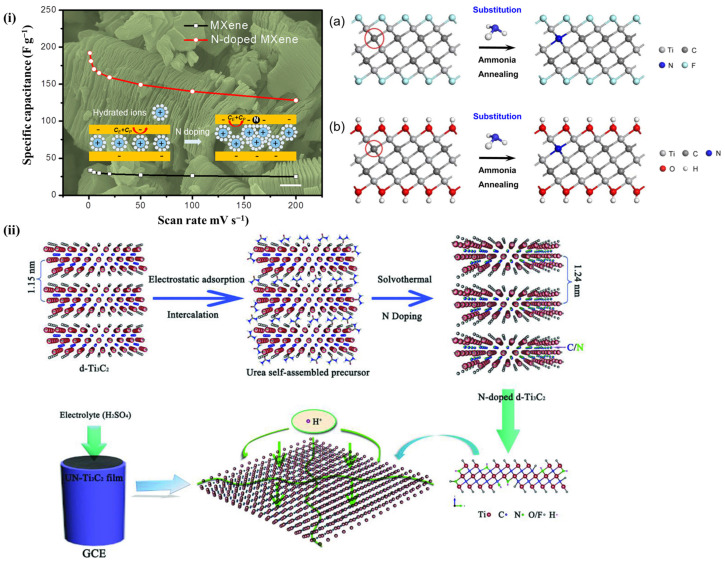
(**i**) An exemplification of charge storage and film-electrode fabrication of UN-Ti_3_C_2_ [96]. (**ii**) Schematic illustration of film preparation and electrode fabrication of the UN-Ti3C2.

**Figure 16 nanomaterials-13-00919-f016:**
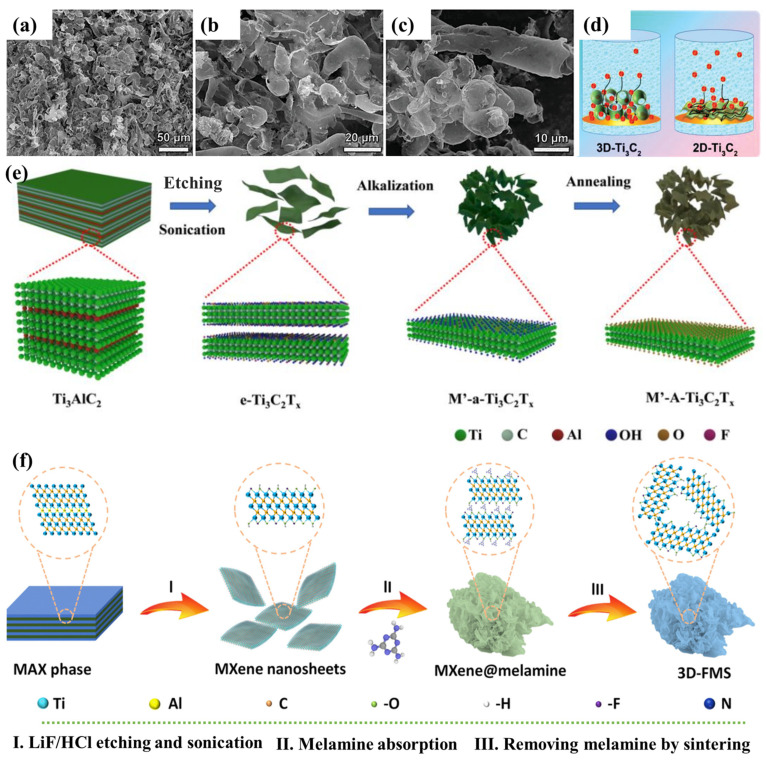
(**a**,**b**) SEM images. (**c**) An exemplification of ion transport in 3D and 2D Ti_3_C_2_ electrodes [125]. (**d**) An exemplification of fabrication [126] and (**e**,**f**) the synthesis of 3D-FMS [127].

**Figure 17 nanomaterials-13-00919-f017:**
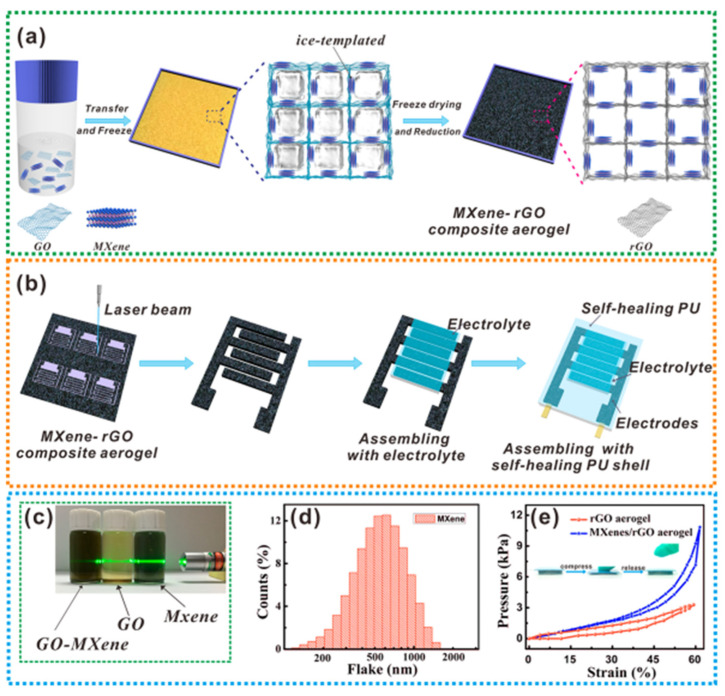
An exemplification of fabrication of (**a**) aerogel, (**b**) MSCs, (**c**) photo of NCs in dispersion of water, (**d**) size of MXene flake, and (**e**) pressure-strain curve [134].

**Figure 18 nanomaterials-13-00919-f018:**
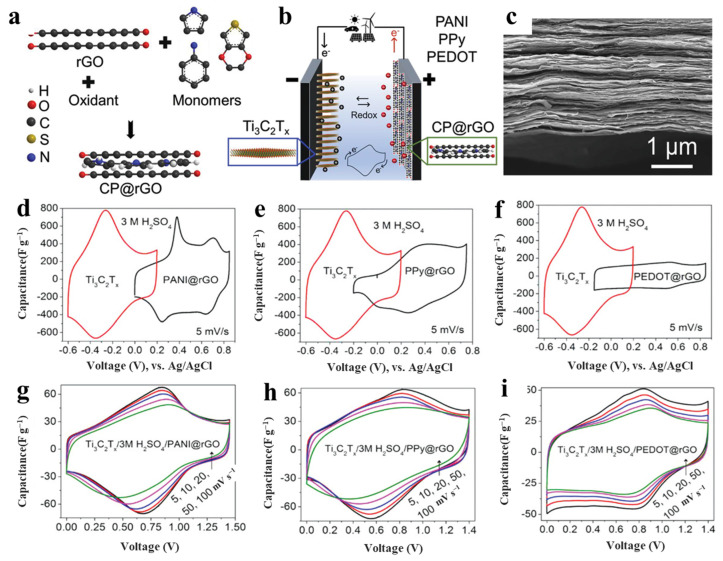
(**a**) An exemplification of MXene synthesis, (**b**) schematic illustration of the tests, (**c**) cross-sectional image and the (**d**–**i**) CV curve of the MXene-based electrodes [135].

**Figure 19 nanomaterials-13-00919-f019:**
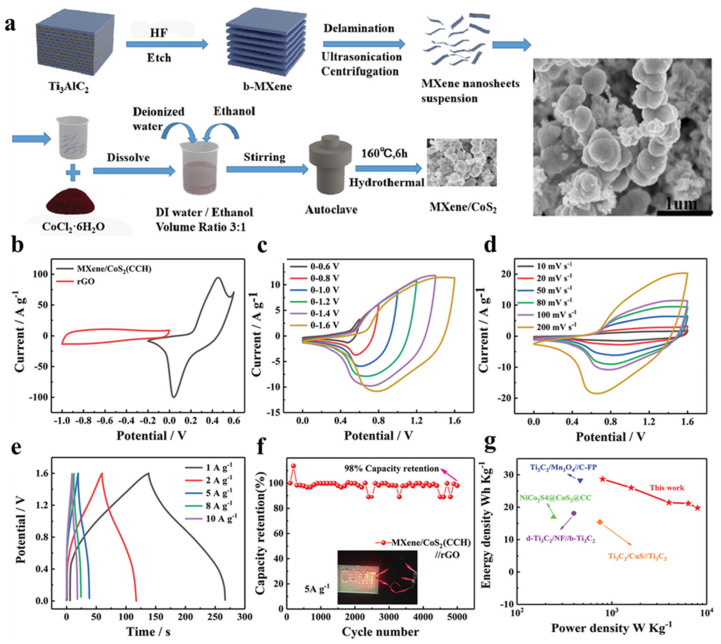
(**a**) Schematic diagram for the synthesis process of MXene/CoS_2_ (CCH) composite with the (**b**) SEM image, and the (**b**–**g**) electrochemical performance of MXene/CoS_2_ (CCH)//rGO ASCs in 2 m KOH aqueous electrolyte [148].

**Figure 20 nanomaterials-13-00919-f020:**
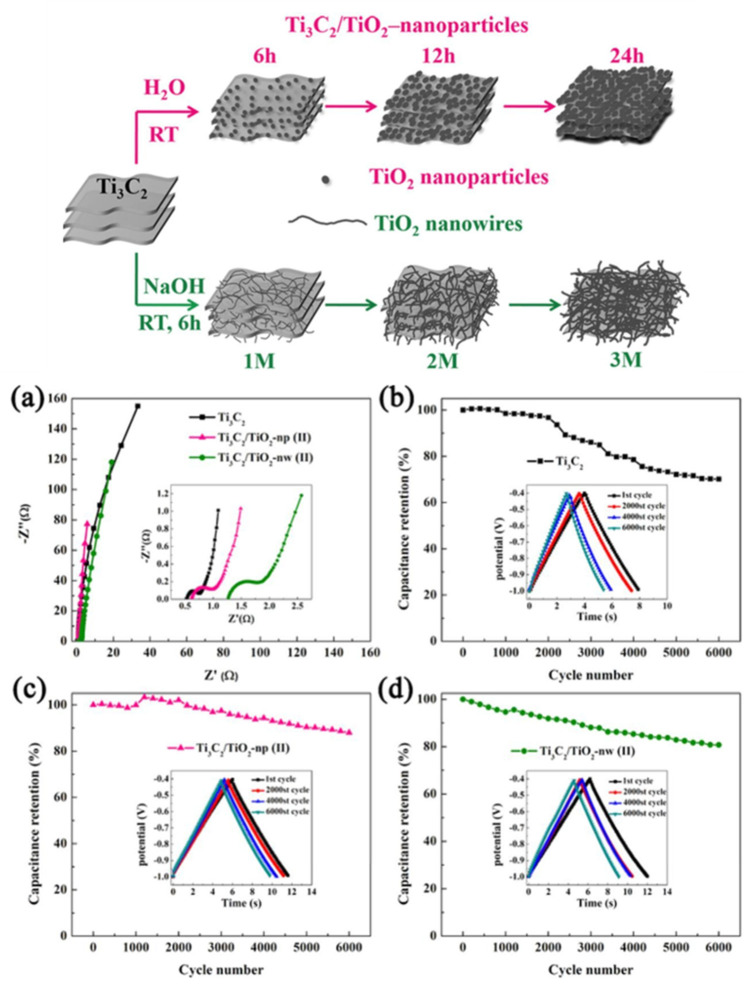
An exemplification of Ti_3_C_2_/TiO_2_ composites fabrication, (**a**) EIS, and (**b**–**d**) cyclability retention of various Ti_3_C_2_/TiO_2_ -electrode [147].

**Figure 21 nanomaterials-13-00919-f021:**
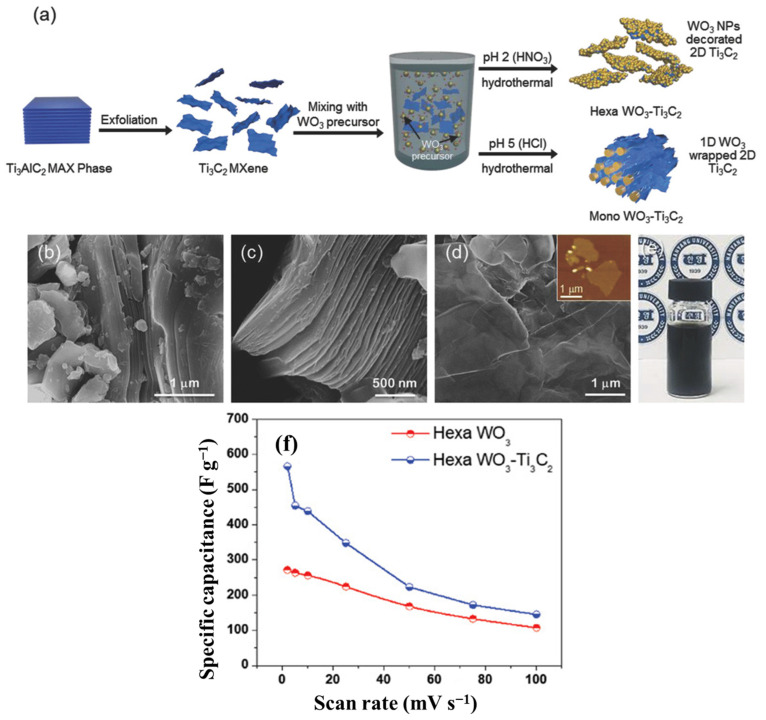
(**a**) An exemplification of fabrication, (**b**–**e**) FESEM, AFM, and vial images with the (**f**) specific capacitance rate of Hexa-WO_3_ electrodes [149].

**Figure 22 nanomaterials-13-00919-f022:**
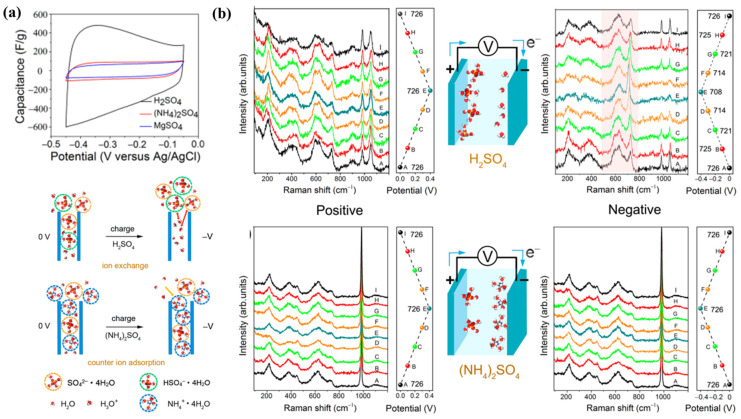
(**a**) CV-curves (**b**) In situ Raman spectra of Ti_3_C_2_T_*x*_ MXene recorded on a positive electrode and a negative electrode [152].

**Figure 23 nanomaterials-13-00919-f023:**
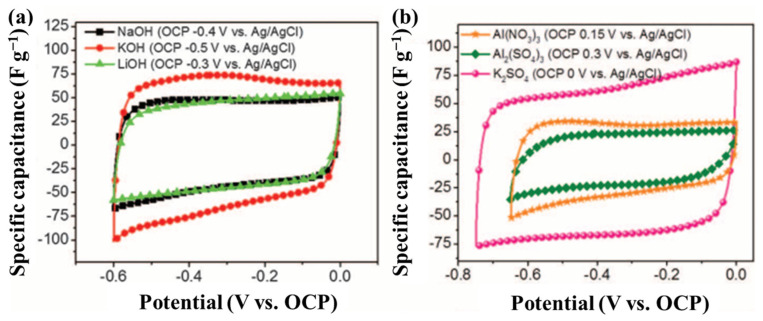
(**a**,**b**) CV profiles of Ti_3_C_2_T*_x_*-based SCs in basic and neutral electrolytes [38].

**Figure 24 nanomaterials-13-00919-f024:**
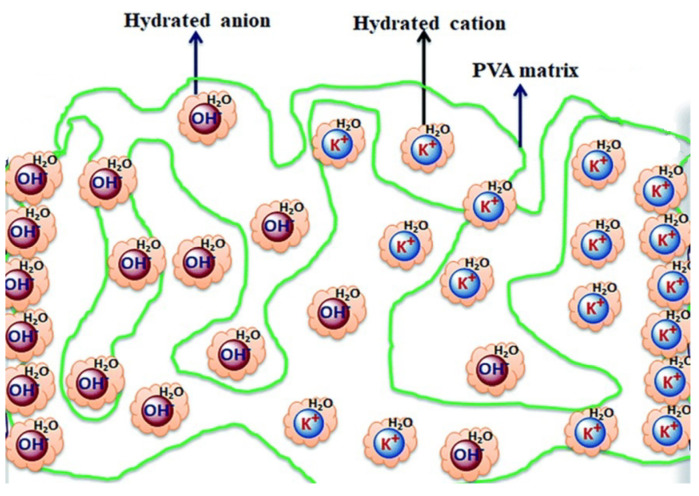
Graphical representation of the hydrogel polymer electrolyte (PVA/KOH/H_2_O) [161].

**Table 1 nanomaterials-13-00919-t001:** Capacitive capabilities of different MXene electrode materials for SCs.

Versatility	Synthesis	Electrode Assembly	Electrolyte	Capacitance	Cycle Stability	Ref.
Ti_3_C_2_T_*x*_	HF etching Ti_3_AlC_2_ and DMSO delamination	Filtrating into film	1 M KOH	350 F cm^−3^	no degradation (10,000 cycles)	[41]
Ti_3_C_2_T_*x*_	HCl/HF etching Ti_3_AlC_2_	Rolling	1 M H_2_SO_4_	900 F cm^−3^ or 245 F g^−1^	no degradation (10,000 cycles)	[39]
Ti_3_C_2_T_*x*_	Lewis acidic molten salt etching Ti_3_SiC_2_	Rolling	1 M LiPF_6_- EC/DMC	738 C g^−1^ or 205 mAh g^−1^	90% (2400 cycles)	[31]
Ti_3_CT_*x*_	HF etching Ti_2_AlC	Rolling	1 M KOH	517 F cm^−3^	no degradation (3000 cycles)	[85]
Ti_3_NT_*x*_	oxygen-assisted molten salt etching Ti_2_AlN_2_ and HCl treatment	Coating	1 M MgSO_4_	201 F g^−1^	140% (1000 cycles)	[88]
V_2_C	HF etching V_2_AlC and TMAOH delamination	Rolling	1 M H_2_SO_4_	487 F g^−1^	83% (10,000 cycles)	[89]
V_4_C_3_T_*x*_	HF etching V_4_AlC_3_	Coating	1 M H_2_SO_4_	~209 F g^−1^	97.23% (10,000 cycles)	[90]
Nb_4_C_3_T_*x*_	HF etching Nb_4_AlC_3_ and TMAOH delamination	Rolling	1 M H_2_SO_4_	1075 F cm^−3^	76% (5000 cycles)	[91]
Ta_4_C_3_	HF etching Ta_4_AlC_3_	Coating	0.1 M H_2_SO_4_	481 F g^−1^	89% (2000 cycles)	[92]
Mo_2_CT_*x*_	HF etching Mo_2_Ga_2_C and TBAOH delamination	Filtrating into film	1 M H_2_SO_4_	700 F cm^−3^	no degradation (10,000 cycles)	[36]
Mo_2_TiC_2_T_*x*_	HF etching and DMSO delamination Mo_2_TiAlC_2_	Rolling	1 M H_2_SO_4_	413 F cm^−3^	no degradation (10,000 cycles)	[43]

**Table 3 nanomaterials-13-00919-t003:** Comparison study of pure MXene SCs, MXene composite SCs, and other SCs.

Electrode	Electrolyte	Gravimetric Capacitance (F g^−1^)	Cycling Stability	Flexibility	Ref.
Ti_3_C_2_T_*x*_	1 M H_2_SO_4_	314 (2 mV s^−1^)	89.1% retention (5 mA g^−1^), 10,000 cycles	-	[52]
Ti_3_C_2_T_*x*_	3 M H_2_SO_4_	380 (10 mV s^−1^)	Over 90% retention (10 mA g^−1^), 10,000 cycles	Flexible film	[78]
Ti_3_C_2_T_*x*_	1 M KOH	130 (2 mV s^−1^)	100% retention (1 mA g^−1^), 10,000 cycles	Flexible film	[162]
Ti_3_C_2_T_*x*_/rGO	3 M H_2_SO_4_	335 (2 mV s^−1^)	100% retention (1 mA g^−1^), 20,000 cycles	Flexible film	[163]
Ti_3_C_2_T_*x*_/CNT	MgSO_4_	125 (2 mV s^−1^)	100% retention (5 A g^−1^), 10,000 cycles	Flexible film	[112]
Ti_3_C_2_T_*x*_/PPy	1 M H_2_SO_4_	416 (5 mV s^−1^)	92% retention (100 mV s^−1^), 25,000 cycles	Flexible film	[117]
Ti_3_C_2_T_*x*_/NiCo_2_S_4_	3 M KOH	1147.47 (1 A g^−1^)	91.1% retention (10 A g^−1^), 3000 cycles	-	[164]
Ti_3_C_2_T_*x*_/graphene	H_2_SO_4_	542 (5 mV s^−1^)	~ 52% retention (1000 mV s^−1^) 5000 cycles	-	[165]
Ti_3_C_2_T_*x*_/hydrogel	H_2_SO_4_	370~165 (5~1000 A g^−1^)	~ 98% retention (10 A g^−1^) 10,000 cycles	-	[166]
CuO@ AuPd@MnO_2_ Core-shell whiskers	1 M KOH	1376 (5 mV s^−1^)	99% retention (5 mVs^−1^) 5000 cycles	-	[167]
Ni-MOF	3 M KOH	988 (1.4 A g^−1^)	96.5% retention(1.4 A g^−1^) 5000 cycles	Flexible film	[168]
PANI-ZIF67	3 M KCl	2146 (10 mV s^−1^)	-	Flexible	[169]
α-Fe_2_O_3_@ C	1 M Na_2_SO_4_	1232.4 (2 mA cm^−2^)	97.6% retention(2 mA cm^−2^) 4000 cycles	Flexible (97.1 6% retention after 4000 bending cycles)	[170]
MOFC/CNT	6 M KOH	381.2 (5 mV s^−1^)	95% retention (5 mV s^−1^) 10,000 cycles	Flexible	[171]
TiO_2_ nanospindles	6 M KOH	897 (0.21 A g^−1^)	75% retention (0.21 A g^−1^) 5000 cycles	-	[172]
NiCo_2_O_4_ NWARs/PPy	3 M KOH	2244 (1 A g^−1^)	89.2% retention (1 A g^−1^) 5000 cycles	-	[173]
CoO-NiO-ZnO	3 M KOH	2115.5 (1 A g^−1^)	87.9% retention (1 A g^−1^) 5000 cycles	-	[174]
ZIF/PPy	1 M Na_2_SO_4_	554.4 (0.5 A g^−1^)	90.7% retention (0.5 A g^−1^) 10,000 cycles	Little capacitance fading up to 180 bending cycles	[175]
UiO-66/polypyrrole	3 M KCl	90.5 (5 mV s^−1^)	96% retention (5 mV s^−1^) 1000 cycles	96% retention after 1000 bending cycles	[176]
PANI/UiO-66	PVA/H_2_SO_4_	1015 (1 A g^−1^)	84% retention (1 A g^−1^) 3500 cycles	90% retention after 800 bending cycle is	[177]

## Data Availability

Data can be made available upon written request to the corresponding author and with a proper justification.

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
