# Peer review of "Recent Advances in Two-Dimensional MXene for Supercapacitor Applications: Progress, Challenges, and Perspectives"

_nanomaterials, 2023, doi:10.3390/nano13050919_

Round 1

Reviewer 1 Report

This review paper reports the recent advances in 2D MXene for supercapacitor applications: progress, challenges and perspectives. This is a research field that is currently a hot issue and is of interest to the readers. However, some minor revisions are needed before being accepted for publication. Below please find the comments in detail. 

1)  As a review paper, this manuscript needs description of important points and in-depth discussion of data, and more published citations are required.

2)  In particular, 2-3 papers citations published in “Nanomaterials” are needed in the discussion section.

3)   It is required to describe a solid electrolyte in the "9. Capacitive Mechanism of MXene in Electrolytes" part.

4)Reference [104] is missing in the text, which may lead to confusion of reference number.

5)  On page 20 of 39 and 21 of 39, the section numbers 7.8 and 7.9 should be wrong.

Author Response

This review paper reports the recent advances in 2D MXene for supercapacitor applications: progress, challenges and perspectives. This is a research field that is currently a hot issue and is of interest to the readers. However, some minor revisions are needed before being accepted for publication. Below please find the comments in detail. 

1)  As a review paper, this manuscript needs description of important points and in-depth discussion of data, and more published citations are required.

  • Thank you for your comment. We revised the review paper and highlighted the revised parts in review paper.

2)  In particular, 2-3 papers citations published in “Nanomaterials” are needed in the discussion section.

  • Thank you for your comment. We added the paper citation in the review paper.

“In addition, the capacitance of the MXene-based electrode in SC can be enhanced by controlling the composition of MXene-containing NCs. For example, the Ni-MOF/Ti3C2Tx nanocomposite has high specific capacitance of 497.6 F g−1 at 0.5 A g−1 in KOH/PVA electrolyte [75], Ti3C2Tx-AuNPs film has capacitance of 696.67 F g−1 at 5 mVs−1 in 3 M H2SO4 electrolyte [76], and MnO2@V2C-MXene had a capacitance approximately 551.8 F g−1, has a retentivity of about 96.5 % after 5000 cycles [77]. These results offered an efficient approach to fabricating high-performance metal oxide-based symmetric capacitors is proposed, and a simple and easy way to improve the design of MXene-based electrodes is presented.”

[75] Li S, Wang Y, Li Y, Xu J, Li T, Zhang T. In Situ Growth of Ni-MOF Nanorods Array on Ti3C2Tx Nanosheets for Supercapacitive Electrodes. Nanomaterials. 2023; 13(3), 610.

[76] Mustafa, B.; Lu, W.; Wang, Z.; Lian, F.; Shen, A.; Yang, B.; Yuan, J.; Wu, C.; Liu, Y.; Hu, W.; Wang, L.; Yu, G. Ultrahigh Energy and Power Densities of d-MXene-Based Symmetric Supercapacitors. Nanomaterials 2022, 12, 3294.

[77] Fatima, M.; Zahra, S.A.; Khan, S.A.; Akinwande, D.; Minár, J.; Rizwan, S. Experimental and Computational Analysis of MnO2@V2C-MXene for Enhanced Energy Storage. Nanomaterials 2021, 11, 1707.305.

3)   It is required to describe a solid electrolyte in the "9. Capacitive Mechanism of MXene in Electrolytes" part.

  • Thank you for your insight. We added the description of solid-state electrolyte in the review paper and highlighted it.

“Solid-state electrolyte (SSE) based SCs have recently attracted considerable interest due to the rapidly increasing power demands of wearable electronics, portable electronics, printable electronics, microelectronics, and highly flexible electronic devices. Electrolytes are essential components of supercapacitors and play an influential role in transferring and balancing the charge between the two electrodes. It is true that the interaction between electrolyte and electrode in electrochemical processes broadly affects the state of the electrode-electrolyte interface and the internal structure of the active material, and it is noteworthy for the advanced application of flexible supercapacitors. Designing MXene materials and suitable solid electrolyte/MXene interfaces is a highly valued approach. The SC that composed of aqueous electrolyte  ionic liquids is available to operate at high voltage and possess high conductivity and capacitance, but it has leakage problem. The use of SSE may help to avoid from leakage problem [166]. SSE is composed of inorganic-solid and organic-polymer electrolyte. The inorganic-solid electrolytes including a single crystal, amorphous compound and polycrystalline, whereas, an organic-polymer solid electrolytes composed of organic-polymer matrix and a salt. The most SSE for SCs have been based on polymer, which is including the solid-polymer, gel-polymer (quasi-solid state) and polyelectrolytes. Among these three SSEs, the gel-polymer electrolytes have been exceptionally sued in SCs due to its high ionic conductivity which derived from liquid-phase. For the preparation of gel-polymer electrolytes, various types of polymer used it such as PVA, PMMA, PVDF-HFP, etc., and water , some organic solvents (DMF, EC, PC) used as a plasticizer [167-170]. Figure 24 represents the graphical representation of the hydro-gel polymer that composed of polymer host (PEO, PVA or PEG) and an aqueous electrolyte (H2SO4, KOH. etc.,) or a conducing salt dissolved in a solvent [171].”

Figure 24. Graphical representation of the hydro-gel polymer electrolyte (PVA/KOH/H2O) [171].

[166] B. E. Francisco , C. M. Jones , S.-H. Lee and C. R. Stoldt , Nanostructured all-solid-state supercapacitor based on Li2S-P2S5 glass-ceramic electrolyte, Appl. Phys. Lett., 2012, 100 , 103902.

[167] J. Duay , E. Gillette , R. Liu and S. B. Lee , Highly flexible pseudocapacitor based on freestanding heterogeneous MnO2/conductive polymernanowire arrays, Phys. Chem. Chem. Phys., 2012, 14, 3329-3337.

[168] C. W. Huang , C. A. Wu , S. S. Hou , P. L. Kuo , C. Te Hsieh and H. Teng , Gel Electrolyte Derived from Poly(ethylene glycol) Blending Poly(acrylonitrile) Applicable to Roll-to-Roll Assembly of Electric Double Layer Capacitors, Adv. Funct. Mater., 2012, 22, 4677-4685.

[169] Y. N. Sudhakar , M. Selvakumar and D. K. Bhat , LiClO4-doped plasticized chitosan and poly(ethylene glycol) blend as biodegradable polymer electrolyte for supercapacitors, Ionics, 2013, 19, 277-285.

[170] C. Ramasamy , J. Palma Del Vel and M. Anderson , An activated carbon supercapacitor analysis by using a gel electrolyte of sodium salt-polyethylene oxide in an organic mixture solvent, J. Solid State Electrochem., 2014, 18, 2217-2223.

[171] B. Pal , A. Yasin , R. Kunwar , S. Yang , M. M. Yusoff and R. Jose , Polymer versus Cation of Gel Polymer Electrolytes in the Charge Storage of Asymmetric Supercapacitors, Ind. Eng. Chem. Res., 2019, 58 , 654-664.

4)Reference [104] is missing in the text, which may lead to confusion of reference number.

  • Thank you for your comment. The reference citations in the review articles were not in the correct order, and the order was corrected because the citation [104] was last in the article.

5)  On page 20 of 39 and 21 of 39, the section numbers 7.8 and 7.9 should be wrong.

  • Thank you for your comment. We revised the sequence of subgroup and highlighted in the review paper.

Reviewer 2 Report

MXene is a type of 2D transition metal carbide and nitride, and its promising energy storage materials highlight the characteristics of high density, high metal-like conductivity, tunable terminals and charge storage mechanisms known as pseudo-alternative capacitance. This review summarizes the current developments, successes, and challenges of MXene in supercapacitors. After carefully reading, it was found that this article needs major revisions because several issues and explanations are still need to be clarified.

1.      The first “2D” that appears in the abstract should be “two-dimensional (2D)”.

2.      Synthesis methods is suggested to be added as a keyword.

3.      To be strict, “capacitor” in the Keywords, in the title of section 3 “Key properties of 2D MXene for capacitor” and other places should be replaced by “supercapacitor”.

4.      “A synthesis approach, various composition problems, material and electrode topology, chemistry, hybridization of MXenes, and other active materials have been reported. ” and “Synthesis approaches, various compositional issues, material and electrode topology, chemistry, and hybridization of MXene with other active materials are also reported.” are repeated in the abstract.

5.      “However, to date, MXenes used in energy storage system applications have rarely been synthesized..”. Really? Please make sure the statement is right.

6.      The language needs to be further polished. Please go through the whole manuscript to polish the language. For example “Electrode materials for SC can be divided into electrical double layer capacitors (EDLCs) and pseudocapacitors…” needs to be revised.

7.      Various nanomaterials including porous carbon materials, transition metal compounds and conducting polymers have been developed as electrode materials for supercapacitor. It is suggested to add more references in the introduction for broad readers, e.g. Journal of Bioresources and Bioproducts 2021, 6 (2), 142-151; Journal of Bioresources and Bioproducts 2022, 7 (4), 245-269; Inorganic Chemistry Frontiers 2022, 9, 6108-6123.

8.      Check paragraph formatting to make sure all paragraphs are indented the same way.

9.      Please write MXene in the same way, there are different forms of MXene in the manuscript, MXene and Mxene.

10.   Please pay attention to the writing of units. “Wh/kg”, “Wh kg-1” and other units should be written in the same way.

11.   Check the paragraph order. “6.4” followed by “7.8”.

12.   There is no explanation for why the paper suddenly changed from introduction “2D MXene” to “3D porous MXene”.

13.   The content on how to reshape the face of the latest MXene was missing in paper.

14.   The Figure 7 (ii)(a) should be the same size as the others. Please remove “A” in Figure 7 (i).

15.   There are some grammatical errors in this manuscript.

16.   The manuscript is like a summary of published studies, lacking critical elements from the viewpoints of experts in the field that are needed for a review article. The authors need to provide more discussion on the innovation and significance of this work. Importantly, the authors should provide more discussion on the future perspectives of the presented research in the abstract, introduction, and conclusion to sublimate manuscript.

17.   References are kind of old. More related articles published in the past two years are suggested to be added, e.g. Journal of Bioresources and Bioproducts 2022, 7 (1), 63-72; Diamond and Related Materials 2022, 128, 109247.

Author Response

MXene is a type of 2D transition metal carbide and nitride, and its promising energy storage materials highlight the characteristics of high density, high metal-like conductivity, tunable terminals and charge storage mechanisms known as pseudo-alternative capacitance. This review summarizes the current developments, successes, and challenges of MXene in supercapacitors. After carefully reading, it was found that this article needs major revisions because several issues and explanations are still need to be clarified.

  1. The first “2D” that appears in the abstract should be “two-dimensional (2D)”.
  • Thank you for your comment We revise the word in abstract part.
  1. Synthesis methods is suggested to be added as a keyword.
  • Thank you for your comment. We added the “synthesis method” is added as a keyword and highlighted it in the review paper.
  1. To be strict, “capacitor” in the Keywords, in the title of section 3 “Key properties of 2D MXene for capacitor” and other places should be replaced by “supercapacitor”.
  • Thank you for your comment. We revised the words and highlighted in the review paper.
  1. “A synthesis approach, various composition problems, material and electrode topology, chemistry, hybridization of MXenes, and other active materials have been reported. ” and “Synthesis approaches, various compositional issues, material and electrode topology, chemistry, and hybridization of MXene with other active materials are also reported.”are repeated in the abstract.
  • Thank you for your comment. We revised the abstract and re-written in the review paper.
  1. “However, to date, MXenes used in energy storage system applications have rarely been synthesized..”. Really? Please make sure the statement is right.
  • Thank you for your comment. we revised the sentence and re-written in the manuscript.

“To date, MXenes used in energy storage system applications have been broadly synthesized, and this paper summarizes the current developments, successes, and challenges of using MXenes in supercapacitors.”

  1. The language needs to be further polished. Please go through the whole manuscript to polish the language. For example “Electrode materials for SC can be divided into electrical double layer capacitors (EDLCs) and pseudocapacitors…” needs to be revised.
  • Thank you for your comment. We checked the grammar mistake and re-written the sentences.
  1. Various nanomaterials including porous carbon materials, transition metal compounds and conducting polymers have been developed as electrode materials for supercapacitor. It is suggested to add more references in the introduction for broad readers, e.g. Journal of Bioresources and Bioproducts 2021, 6 (2), 142-151; Journal of Bioresources and Bioproducts 2022, 7 (4), 245-269; Inorganic Chemistry Frontiers 2022, 9, 6108-6123.
  • Thank you for your insight. We cited the above all papers in the review paper.

“In addition, the porous carbon NCs [20, 21] and N-doped porous carbon [22], and the combination of MXene with porous carbon [23], transition-metal compound and MXene/polymer compounds had emerged as a promising candidates for SCs [24].”

[20] B. Yan, L. Feng, J. Zheng, S. Jiang,, C. Zhang, Y. Ding, J. Han, W. Chen, S. He, High performance supercapacitors based on wood-derived thick carbon electrodes synthesized via green activation process, Inorg. Chem. Front., 2022, 9, 6108-6123.

[22] S. Zheng, J. Zhang, H. Deng, Y. Du, X. Shi, Chitin derived nitrogen-doped porous carbons with ultrahigh specific surface area and tailored hierarchical porosity for high performance supercapacitors, Journal of Bioresources and Bioproducts, 2021, 6 (2), 142-151.

[24] J. Xiao, H. Li, H. Zhang, S. He, Q. Zhang, K. Liu, S. Jiang, G. Duan, K. Zhang, Nanocellulose and its derived composite electrodes toward supercapacitors: Fabrication, properties, and challenges, Journal of Bioresources and Bioproducts, 2022, 7 (4), 245-269.

  1. Check paragraph formatting to make sure all paragraphs are indented the same way.
  • Thank you for your comment. We revised the format of the paragraphs.
  1. Please write MXene in the same way, there are different forms of MXene in the manuscript, MXene and Mxene.
  • Thank you for your comment. We formatted the “MXene” words in the review paper and highlighted.
  1. Please pay attention to the writing of units. “Wh/kg”, “Wh kg-1” and other units should be written in the same way.
  • Thank you for your comment. We check the unit format in the review paper.
  1. Check the paragraph order. “6.4” followed by “7.8”.
  • Thank for your comment. We checked the numbering order of the paragraphs in a review paper.
  1. There is no explanation for why the paper suddenly changed from introduction “2D MXene” to “3D porous MXene”.
  • Thank you for your insight. We added the properties of 3D porous MXene in introduction part, and highlighted it in the review paper.

“MXene materials used in SC generally belong mainly to the 2D structure, and from a structural point of view, the horizontal aggregation and re-stacking of MXene nanosheets due to strong van der Waals interactions between adjacent layers led to the accessibility of electrolyte ions and the use of the entire 2D MXene surface limits these possibilities. One advanced approach to overcome this limitation has been proven to be the design of an open structure of MXene nanosheets, which offers advantages such as increasing the electrochemically accessible surface area of MXenes and improving the rate of ion transport to active redox sites by tailoring the properties or morphology. Various strategies to tune the morphology, such as particle size control, interlayer spacing expansion, three-dimensional (3D) porous structures, and vertical line design, have been proposed in research publications for high-capacity and power-performance MXenes. Designing a 3D/porous electrode structure with a large active surface area accessible to ions and pores connected between ion transport channels can more effectively improve the high-rate capability of MXene. Various methods, such as hard-templating strategy [29], different types of freeze-drying methods [30, 31], chemical-cross linking [32], oxidative etching [33], and self-assembly methods [34], have been used for the preparation of 3D porous MXene materials, most of the 3D porous MXene material had mesopore / macropore with wide-size distribution profiles.”

[29] M.Q. Zhao, X. Xie, C.E. Ren, T. Makaryan, B. Anasori, G. Wang, Y. Gogotsi, Hollow MXene Spheres and 3D Macroporous MXene Frameworks for Na-Ion Storage, Adv. Mater. 2017, 29, 1702410.

[30] J. Liu, H.-B. Zhang, X. Xie, R. Yang, Z. Liu, Y. Liu, Z.Z. Yu, Multifunctional, Superelastic, and Lightweight MXene/Polyimide Aerogels. Small, 2018, 1802479.

[31] X. Li, X. Yin, H. Xu, M. Han, M. Li, S. Liang, L. Cheng, L. Zhang, Ultralight MXene-Coated, Interconnected SiCnws Three-Dimensional Lamellar Foams for Efficient Microwave Absorption in the X-Band, ACS Appl. Mater. Interfaces 2018, 10, 34524–34533.

[32] C. Xing, S. Chen, X. Liang, Q. Liu, M. Qu, Q. Zou, J. Li, H. Tan, L. Liu, D. Fan, H. Zhang, Two-Dimensional MXene (Ti3C2)-Integrated Cellulose Hydrogels: Toward Smart Three-Dimensional Network Nanoplatforms Exhibiting Light-Induced Swelling and Bimodal Photothermal/Chemotherapy Anticancer Activity, ACS Appl. Mater. Interfaces 2018, 10, 27631–27643.

[33] C.E. Ren, M.Q. Zhao, T. Makaryan, J. Halim, M. Boota, S. Kota, B. Anasori, M.W. Barsoum, Y. Gogotsi, Porous Two-Dimensional Transition Metal Carbide (MXene) Flakes for High-Performance Li-Ion Storage, ChemElectroChem 2016, 3, 689–693.

[34] Y.T. Liu, P. Zhang, N. Sun, B. Anasori, Q.-Z. Zhu, H. Liu, Y. Gogotsi, B. Xu, Self-Assembly of Transition Metal Oxide Nanostructures on MXene Nanosheets for Fast and Stable Lithium Storage, Adv. Mater. 2018, 30, 1707334.

  1. The content on how to reshape the face of the latest MXene was missing in paper.
  • Thank you for your comment. We added the different synthesis methods of MXene in “2. Synthesis of MXene 2D” part and highlighted it in the review paper.

“In 2018, an F-free hydrothermal alkali etching method was used for the synthesis of MXenes [52]. Therefore, MXene formation from MAX (Ti3SiC2) with non-Al was reported using a hybrid HF/H2O2 etchant solution [53]. Since then, different synthesis studies have been reported in 2019, such as adopting MXene-metal oxides, MXene-polymers, and MXene-carbon NCs, representing the composite MXene as a state-of-the-art hybrid material for versatile applications [54-56]. In 2016, Urbankowski et al. proposed a short period and effective etching method, by using a molten fluorine-containing salt mixture of LiF, NaF, and KF as an etchant to obtain the first two-dimensional transition metal nitride MXene [57].

In addition to fluorine-containing salt melts, Lewis acid melt salts have proven to be effective tools for etching the MAX phase. The “A” atom in the MAX precursor, and the cation in the Lewis acid, are removed by an oxidation-reduction reaction. In 2020, Talapin et al. proposed a general strategy and synthesized a variety of MXenes with single surface terminations (O, S, Se, Cl, Br, NH, and Te) as well as bare MXene without surface termination and found that some of the Nb2C series MXenes, including Nb2CCl MXene, have superconductivity under low-temperature conditions [58]. The above-mentioned etching methods were based on removing the “A”-layer from MAX and forming MXene with and without surface terminations, and the lateral dimension of the chemically derived MXene was between several hundreds of nanometers to ∼10 μm. Three different syntheses, chemical vapor deposition (CVD), lithiation −expansion−micro explosion mechanism, and in-situ electrochemical synthesis methods, are listed in the bottom-up approach used for MXene synthesis. In 2015, Xu et al. proposed a CVD method for synthesizing ultrathin and large-scale MXenes (Mo2C) [59]. Two years later, another research group reported a CVD method for MXene, which had a lateral dimension at the centimeter level [60]. In 2020, Buke et al. studied the influence of impurities on Mo2C crystal formation [61]. For the lithiation-expansion-micro explosion mechanism, Sun et al. developed a new method for preparing single-layer or few-layer MXenes from Ti3AlC2 MAX. In addition to that, the Ti3SiC2 MAX was used as a precursor material in this method (in 2019) [62]. Recently, in 2020, Zhi et al. reported an integrated process that combines the in situ etching of MAX and ion storage of MXene using LiTFSI and Zn(OTF)2-mixed ionic electrolyte as the etchant solution [63].”

[52] T. Li, L. Yao, Q. Liu, J. Gu, R. Luo, J. Li, X. Yan, W. Wang, P. Liu, B. Chen, W. Zhang, W. Abbas, R. Naz, D. Zhang, Fluorine-free synthesis of high-purity Ti3C2Tx (T=OH, O) via Alkali treatment. Angew. Chem. Int. Ed. Engl. 2018, 57, 6115–6119. 

[53] M. Alhabeb, K. Maleski, T. S. Mathis, A. Sarycheva, C. B. Hatter, S. Uzun, A. Levitt, Y. Gogotsi, Selective etching of silicon from Ti3SiC2 (MAX) to obtain 2D titanium carbide (MXene) Angew. Chem. Int. Ed. Engl. 2018, 57, 5444–5448.

[54] G. S. Gund, J. H. Park, R. Harpalsinh, M. Kota, J. H. Shin, T. I. Kim, Y. Gogotsi, H. S. Park, MXene/polymer hybrid materials for flexible AC-filtering electrochemical capacitors. Joule2019, 3, 164–176.

[55] J. Li, A. Levitt, N. Kurra, K. Juan, N. Noriega, X. Xiao, X. Wang, H. Wang, H. N. Alshareef, Y. Gogotsi, MXene-conducting polymer electrochromic microsupercapacitors. Energy Storage Mater. 2019, 20, 455–461.

[56] T. Shang, Z. Lin, C. Qi, X. Liu, P. Li, Y. Tao, Z. Wu, D. Li, P. Simon, Q. Yang, 3D macroscopic architectures from self-assembled MXene hydrogels. Adv. Funct. Mater. 2019, 29. 1903960.

[57] P. Urbankowski, B. Anasori, T. Makaryan, D. Er, S. Kota, P. L. Walsh, M. Zhao, V. B. Shenoy, M. W. Barsoum, Y. Gogotsi, Synthesis of two-dimensional titanium nitride Ti4N3 (MXene). Nanoscale. 2016, 8 (22), 11385- 11391.

[58] V. Kamysbayev, A. S. Filatov, H. C. Hu, X. Rui, F. Lagunas, D. Wang, R. F. Klie, Covalent surface modifications and superconductivity of two-dimensional metal carbide MXenes. Science. 2020, 369 (6506), 979-983.

[59] C. Xu, L. Wang, Z. Liu, L. Chen, J. Guo, N. Kang, X. L. Ma, H. M. Cheng, W. Ren, Large-area high-quality 2D ultrathin Mo2C superconducting crystals. Nat Mater. 2015, 14 (11), 1135-1141.

[60] D. Geng, X. Zhao, Z. Chen, W. Sun, W. Fu, J. Chen, W. Liu, W. Zhou, K. P. Loh, Direct synthesis of large-area 2D Mo2C on in situ grown graphene. Adv Mater. 2017, 29 (35), 1700072.

[61] F. Turker, O. R. Caylan, N. Mehmood, T. S. Kasirga, C. Sevik, C. G. Buke, CVD synthesis and characterization of thin Mo 2 C crystals. Journal of the American Ceramic Society. 2020,103 (10), 5586–5593.

[62] Z. Sun, M. Yuan, L. Lin,  H. Yang, C. Nan, H. Li, G. Sun, X. Yang, Selective lithiation–expansion–microexplosion synthesis of two-dimensional fluoride-free Mxene. ACS Mater Lett. 2019, 1 (6), 628-632.

[63] Z. Li, Y. Ren, L. Mo, C. Liu, K. Hsu, Y. Ding, X. Zhang, X. Li, L. Hu, D. Ji, G. Cao, Impacts of oxygen vacancies on zinc ion intercalation in VO2. ACS Nano. 2020, 14 (5), 5581-5589.

  1. The Figure 7 (ii)(a) should be the same size as the others. Please remove “A” in Figure 7 (i).
  • Thank you for your comment. We revised the Figure 7 and re-attached in the review paper.
  1. There are some grammatical errors in this manuscript.
  • Thank you for your insight. We have checked for grammatical errors in the entire review paper.
  1. The manuscript is like a summary of published studies, lacking critical elements from the viewpoints of experts in the field that are needed for a review article. The authors need to provide more discussion on the innovation and significance of this work. Importantly, the authors should provide more discussion on the future perspectives of the presented research in the abstract, introduction, and conclusion to sublimate manuscript.
  • Thank you for your comment. We revised the whole review paper.
  1. References are kind of old. More related articles published in the past two years are suggested to be added, e.g. Journal of Bioresources and Bioproducts 2022, 7 (1), 63-72; Diamond and Related Materials 2022, 128, 109247.
  • Thank you for your comment. We added the above papers and revised part is highlighted in the review paper.

[21] J. Zheng, B. Yan, Q. Zhang, C. Zhang, W. Yang, J. Han, S. Jiang, S. He, Potassium citrate assisted synthesis of hierarchical porous carbon materials for high performance supercapacitors, Diam. Relat. Mater. 2022, 128, 109247.

[23] L. Wei, W. Deng, S. Li, Z. Wu, J. Cai, J. Luo, Sandwich-like chitosan porous carbon Spheres/MXene composite with high specific capacitance and rate performance for supercapacitors, Journal of Bioresources and Bioproducts, 2022, 7 (1), 63-72.

Reviewer 3 Report

The topic of the proposed review is a very important one. The authors did a good job of assembling a very extensive set of literature on Mxene for supercapacitor applications.

However, the manuscript is poorly written. The English is poor. The Introduction is not so much an introduction to the field as a direct starting discussion of some of the topics. We also believe that, if the topic is of such importance, then the apparent lack of publications dating later than 2021 is puzzling.

Author Response

The topic of the proposed review is a very important one. The authors did a good job of assembling a very extensive set of literature on Mxene for supercapacitor applications. However, the manuscript is poorly written. The English is poor. The Introduction is not so much an introduction to the field as a direct starting discussion of some of the topics. We also believe that, if the topic is of such importance, then the apparent lack of publications dating later than 2021 is puzzling.

  • Thank you for your comment. We revised the introduction, other parts and the reference in the review paper.

Reviewer 4 Report

In this review article, the authors have summarized the current research progress in MXene based materials with particular emphasis on Ti based materials for the application in supercapacitors. The article is well written and well organized, authors have discussed the synthesis methods, important properties of MXene for application in supercapacitors. Article also discusses the methods for modification of  MXenes compositions and electrode structures to enhance the supercapacitor performance. The article will serve as a good reference for the scientific communities working on the development of MXene or other 2-D materials for application in supercapacitors or energy storage devices. Reviewer recommends the publication of the article with following suggestions

·         Please, highlight in the introduction how the present review is different from the other review articles on MXene based materials for supercapacitors, for eg.

o   https://onlinelibrary.wiley.com/doi/full/10.1002/smm2.1130

o   https://pubs.acs.org/doi/10.1021/acs.energyfuels.1c04104

o   https://www.sciencedirect.com/science/article/pii/S2211339821000423  

o   https://www.mdpi.com/2073-4352/12/8/1099

·         Recommended to add a device architecture of supercapacitor for eg. near section 4 or 6.

·         Please check the labelling of section 7 and its sub sections

·         If possible, please add a section on the stability/lifetime of the  Mxenes based supercapacitors, and add comparative comments  w.r.t other materials widely used or investigated

Author Response

In this review article, the authors have summarized the current research progress in MXene based materials with particular emphasis on Ti based materials for the application in supercapacitors. The article is well written and well organized, authors have discussed the synthesis methods, important properties of MXene for application in supercapacitors. Article also discusses the methods for modification of  MXenes compositions and electrode structures to enhance the supercapacitor performance. The article will serve as a good reference for the scientific communities working on the development of MXene or other 2-D materials for application in supercapacitors or energy storage devices. Reviewer recommends the publication of the article with following suggestions

  • Please, highlight in the introduction how the present review is different from the other review articles on MXene based materials for supercapacitors, for e.g.

o   https://onlinelibrary.wiley.com/doi/full/10.1002/smm2.1130

o   https://pubs.acs.org/doi/10.1021/acs.energyfuels.1c04104

o   https://www.sciencedirect.com/science/article/pii/S2211339821000423  

o   https://www.mdpi.com/2073-4352/12/8/1099

  • Thank you for your comment. MXene-based NCs have emerged as promising candidates for SCs with high performance due to their feature properties. In this review paper, we strived to summarize the fundamental theories, research results, and the new ways researchers have adopted in recent years to design high-performance SCs. Compared with other revie paper, briefly and concisely, the first part of this publication summarizes the synthesis method, together with the research results of the main properties that influence the use of 2D MXene in supercapacitors, how the charge storage ability of 2D MXene depends on the surface functional group, and the electrolyte selectivity of MXene- and how it can affect the preparation of electrodes based on. In addition, it is worth mentioning that, given its unique properties, more novel electrodes based on Ti3C2Tx with optimized performance are expected to emerge and the class will move to practical energy storage applications in various fields such as multi-flexible and wearable electronics. We believe that this review paper can contribute to the collection of a very extensive literature on MXene for supercapacitor applications.
  • Recommended to add a device architecture of supercapacitor for eg. near section 4 or 6.
  • Thank you for your comments. We have added a summary graph of the fabrication methods of MXene-based supercapacitors.

“Figure 9 shows the different patterning methods that are used for MXene-based SCs. Generally, the fabrication methods include two classes: a. direct patterning of MXene on the current collector/substrate by laser scribing or reactive ion-etching, b. transferring the MXene ink into different patterns by following printing methods. In addition to these methods, various types of methods have been used for the preparation of MXene-based SCs, such as spray coating with laser cutting, vacuum filtration with laser cutting, direct writing, and freeze-drying with laser cutting methods.”

Round 2

Reviewer 2 Report

The manuscript has been well revised according to the comments and suggest to be accepted.

Reviewer 3 Report

The authors have responded to one of my earlier comments, about adding more recent literature. However, the whole manuscript was NOT rewritten as requested. It reads poorly and the introduction remains inappropriate. A review article is expected to be outstanding in its writing in order to be an important contribution and this manuscript fails completely. This should NOT be published in its current form.